

# Self-sustained Oscillations in the Atmosphere (0 − 110 km) at Long Periods

Dirk Offermann(1), Christoph Kalicinsky(1), Ralf Koppmann(1), and  Johannes Wintel(1,2)

(1)  Institut für Atmosphären-und Umweltforschung, Bergische Universität Wuppertal, Wuppertal, Germany
(2)  Now at  Elementar Analysensysteme GmbH, Langenselbold, Germany

Corresponding author: Dirk Offermann, (offerm@uni-wuppertal.de)

Key Points:  -  multi-decadal oscillations in GCM and measurements (up to 341 yr)
             -  self-sustained oscillations linked to the atmosphere basic dynamics
             -  vertical amplitude and phase structure similar for all oscillation periods

Abstract



Self-generated (self-sustained) oscillations have been observed in measured atmospheric
data at multi-annual periods. These oscillations are also present in General Circulation
Models even if their boundary conditions with respect to solar cycle, sea surface temperature,
and trace gas variability are kept constant. The present analysis contains temperature
oscillations with periods from below 5 yr up to 341 yr in an altitude range from the Earth's
surface to the lower thermosphere (110 km). The periods are quite robust as they are found to
be the same in different model calculations and in atmospheric measurements. The
oscillations show vertical profiles with special structures of amplitudes and phases. They form
layers of high / low amplitudes that are a few dozen km wide. Within the layers the data are
correlated. Adjacent layers are anticorrelated. A vertical displacement mechanism is indicated
with displacement heights of a few 100 metres. Vertical profiles of amplitudes and phases of
the various oscillation periods as well as their displacement heights are surprisingly similar.
The oscillations are related to the thermal and dynamical structure of the middle atmosphere.
These results are from latitudes/longitudes in Central Europe.
Short summary

Atmospheric oscillations with periods up to several 100 years exist at altitudes up to 110
km. They are also seen in computer models (GCM) of the atmospheric. They are often
attributed to external  influences from the sun, from the oceans, or from  atmospheric
constituents. This is difficult to verify as the atmosphere cannot be manipulated in an
experiment. However, a GCM can be changed arbitrarily! Doing so we find that long period
oscillations can be excited internally in the atmosphere.
1    Introduction
Multi-annual oscillations with periods between 2 and 11 years have frequently been discussed
for the atmosphere and the ocean. Major examples are the Quasi-Biennial Oscillation (QBO),



solar cycle related variations near 11 years and 5.5 years, and the El Nino/Southern
Oscillation (ENSO). (For references see for instance Offermann et al., 2015.)
Self-excited oscillations in the ocean of such periods have been described for instance by
White and Liu (2008). Self-excited oscillations in the atmosphere with periods between 2.2
and 5.5 yr have been shown in a large altitude regime by Offermann et al. (2015). Their
periods are surprisingly robust, i.e. there is little change with altitude. They are also present in
general circulation models, the boundaries of which are kept constant.
Oscillations of much longer periods in the atmosphere and the ocean have also been
reported. Biondi et al. (2001) found bi-decadal oscillations in local tree ring records that date
back several centuries. Kalicinsky et al. (2016, 2018) recently presented a temperature
oscillation near the mesopause with a period near 25 years which may be interpreted as a self-
excited oscillation. Low-frequency oscillations (LFO) on local and global scales in the multi-
decadal range (50-80 yr) have been discussed several times (e.g., Schlesinger and Ramankutty
(1994); Minobe (1997); Polyakov et al.(2003); Dai et al.(2015); Dijkstra et al.(2005)). Some
of these results were intensively discussed as internal variability of the atmosphere-ocean
system, for instance as the internal interdecadal modes AMV (Atlantic Multidecadal
Variability) and PDO/IPO (Pacific Decadal Oscillation/Interdecadal Pacific Oscillation) (e.g.
Meehl et al., 2013; 2016; Lu et al., 2014; Deser et al., 2014; Dai et al., 2015.) Multidecadal
variations (40-80 years) of Arctic-wide surface air temperatures were, however, related to
solar variability by Soon (2005). Some of these long period variations have been traced
backwards for two or more centuries (Minobe, 1997; Biondi et al., 2001; Mantua and Hare,
2002; Gray et al., 2004). Multidecadal oscillations have also been discussed extensively as
internal climatic variability in the context of the long term climate change (temperature
increase) in the IPCC AR5 Report (e.g. Flato et al., 2013).
Even longer periods of oscillations in the ocean and the atmosphere have also been
reported. Karnauskas et al. (2012) find centennial variations in three general circulation
models of the ocean. These variations occur in the absence of external forcing, i.e. they show
internal variabilities on the centennial time scale. Internal variability in the ocean on a
centennial scale is also discussed by Latif et al. (2013) on the basis of model simulations.
Measured data of a 500 year quasi-periodic temperature variation are shown by Xu et al.
(2014). They analyze a more than 5000 year long pollen record in East Asia. Very long
periods are found by Paul and Schulz (2002) in a climate model. They obtain internal
oscillations with periods of 1600-2000 years.
All long period oscillations cited here refer to temperatures of the ocean or the land/ocean
system. It is emphasized that on the contrary the self-excited multi-annual oscillations
described by Offermann et al. (2015) and those discussed in the present paper are properties
of the atmosphere, and exist in a large altitude regime between the ground and 110 km
altitude. They are not linked to the ocean.
In the present paper the work of Offermann et al. (2015) is extended to multi-decadal and
centennial periods. Internal oscillations in the atmosphere are studied in three general
circulation models. The analysis is locally constrained (Central Europe), but vertically
extended up to 110 km. The model boundary conditions (sun, ocean, trace gases) are kept
constant. The results of  model runs with HAMMONIA, WACCM, and ECHAM6 were made
available to us. They simulate 34 years, 150 years, and 400 years of atmospheric behavior,
respectively. The corresponding results are compared to each other. Most of the analyses are
performed for atmospheric temperatures.
In Section 2 of this paper the three models are described and the analysis method is
presented. In Section 3 the oscillations obtained from the three models are compared. The
vertical structures of the periods, amplitudes, and phases of the self-sustained oscillations are
described. In Section 4 the results are discussed. Section 5 gives a summary and some
conclusions.



2      Model data and their analysis
2.1   Self-sustained oscillations and their vertical structures
In an earlier paper (Offermann et al., 2015) multi-annual oscillations  with periods of about
2 - 5 years have been described at altitudes up to 110 km. These were found in temperature
data of  HAMMONIA model runs (see below). They were present in the model even if the
model boundary conditions (solar irradiance, sea-surface temperatures and sea ice, boundary
values of green-house gases) were kept constant. Therefore they were interpreted as self-
sustained (self-excited) oscillations. The periods were found to be quite robust as they did not
change much with altitude. Robust periods are typical of  self-excited oscillations (Pikovsky
et al.,2003). The oscillations showed particular vertical structures of amplitudes and phases.
Amplitudes did not increase exponentially with altitude as they do with  atmospheric waves.
They rather varied with altitude between maximum and near zero values in a nearly regular
manner. Phases showed jumps of about 180° at the altitudes of the amplitude minima, and
were about constant in between. There were indications of synchronization of amplitudes and
phases.
The periods analyzed in the earlier paper have been restricted to below 5.5 yr. Much longer
periods have been described in the literature. It is therefore of interest to see whether such
longer periods could also be self-excited in the models.
Figure 1 shows an example of such temperature structures for an oscillation with a period
of 17.3 ±0.8 years obtained from the HAMMONIA model discussed below. This picture is
typical of the oscillations in Offermann et al. (2015) and of the oscillations discussed in the
present paper. The periods at the various altitudes are close to their mean value even though
the error bars are fairly large. There is no indication of systematic altitude variations, and
therefore the mean is taken as a first approximation. At some altitudes the periods could not
be determined. In these cases the periods were prescribed by the mean of the derived periods
(dash-dotted red vertical line, 17.3 yr) to obtain approximate amplitudes and phases at these
altitudes (see Offermann et al., 2015). Details of the derivation of periods, amplitudes, and
phases are given in Section 3.
2.2   HAMMONIA
The HAMMONIA model (Schmidt et al., 2006) is based on the ECHAM5 general circulation
model (Röckner et al.,2006), but extends the domain vertically to $2x10^{-7}$ hPa, and is coupled
to the MOZART3 chemistry scheme (Kinnison et al., 2007). The simulation analyzed here
was run at a spectral resolution of T31 with 119 vertical layers. The relatively high

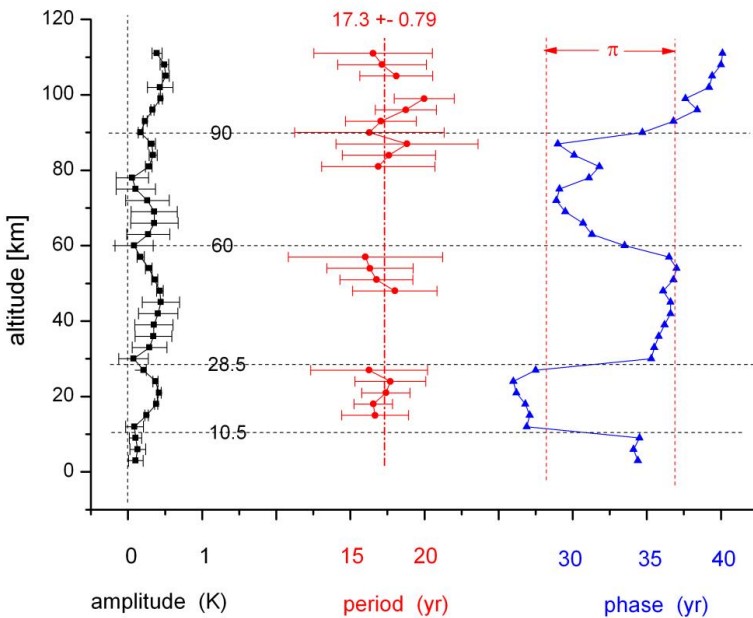

Fig.1     Vertical structures of self-sustained oscillation periods near 17.3 ± 0.8 yr from
HAMMONIA temperatures.
Missing period values could not be derived from the data. They were prescribed as the mean
value 17.3 yr (dash-dotted vertical red line, see text and Section 3.2). Phases are relative
values.

vertical resolution of less than 1 km in the stratosphere allows an internal generation of the
QBO. Here we analyze the simulation (with fixed boundary conditions) that was called "Hhi-
max" in Offermann et al. (2015), but instead of only 11 we use 34 simulated years. Further
details of the simulation are given by Schmidt et al. (2010).
    An example of the HAMMONIA data is given in Fig.2 for 0 km and 3 km altitudes. The
HAMMONIA data were searched for self-sustained oscillations up to 110 km. The detailed
analysis is described below (Section 3.2). Nine oscillations were identified with periods
between 5.3 yr and 28.5 yr. They are listed in Table 2. The oscillation shown in Fig. 1 (17.3
yr) is from about the middle of this range.

2.3 WACCM

    Long runs with chemistry-climate models (CCMs) having restricted boundary conditions
are not frequently available. A model run much longer than 34 years became available from
the CESM-WACCM4 model. This 150 year run was analyzed from the ground up to 108 km.
The model experiments are described in Hansen et al. (2014). Here, the experiment with
monthly varying constant climatological SSTs and sea ice has been used. Other boundary
conditions such as Greenhouse Gases (GHG) and Ozone Depleting Substances (ODP) were
kept constant at 1960 values.



Solar cycle variability, however, was not kept constant during this model experiment.
Spectrally resolved solar irradiance variability as well as variations of the total solar
irradiance and the F10.7cm solar radio flux were used from 1955 to 2004 from Lean et al.
(2005). Thereafter solar variations from 1962-2004 were repeated several times to reach 150
years in total. Details are given in Matthes et al. (2013).
The WACCM data were analyzed for self-excited oscillations in the same manner as the
HAMMONIA data. Here, the emphasis is on longer periods. Besides many shorter
oscillations, nine oscillations with periods of more than 20 years were found. The longest
period is 147 years. These results are included to Table 2.
235    .
2.4   ECHAM6
The longest computer run available to us, covering 400 years, is from ECHAM6. ECHAM6
(Stevens et al., 2013) is the successor of ECHAM5, the base model of HAMMONIA. Major
changes relative to ECHAM5 include an improved representation of radiative transfer in the
solar part of the spectrum, a new description of atmospheric aerosol, and a new representation
of the surface albedo. While the standard configuration of ECHAM5 used a model top at 10
hPa, this was extended to 0.01 hPa in ECHAM6. As the atmospheric component of the Max-
Planck-Institute Earth System Model (MPI-ESM, Giorgetta et al., 2013) it has been used in a
large number of model intercomparison studies related to the Coupled Model Intercomparison
Project phase 5 (CMIP5). The ECHAM6 simulation analyzed here was run at T63 spectral
resolution with 47 vertical layers (not allowing for an internal generation of the QBO). All
boundary conditions were fixed to constant values, taken as an average of the years 1979 to
250    2008.
The temperature data were analyzed as the other data sets described above. Eighteen
oscillation periods longer than 20 yr were obtained (Table 2), with the typical vertical
structures of self-sustained oscillations. The longest period is $341.2 \pm 37.2$ yr
A summary of the model properties is given in Table 1. All analyses in this paper are for
Central Europe. The vertical model profiles are for 50°N, 7°E.
3   Model results
3.1   Vertical correlations of atmospheric temperatures
Figure1  indicates that there are some vertical correlation structures in the atmospheric
temperatures. This was studied in detail for the HAMMONIA and ECHAM6 data.
Ground temperature residues from the HAMMONIA run 38123 (34 years) are shown in
Fig. 2 (black squares). The mean temperature is 281.89 K, which was subtracted from the
model data. The boundary conditions (sun, ocean, green house gases) have been kept constant
, as discussed above. The temperature fluctuations thus show the internal atmospheric
variability (standard deviation is $\sigma = \pm 0.62$ K). This variability is frequently termed
"(climate) noise" in the literature. It will be checked whether this notion is justified in the
present case.
Also shown in Fig. 2 are the corresponding HAMMONIA data for 3 km altitude. The mean
temperature is 266.04 K, the standard deviation is $\sigma = \pm 0.41$ K. The statistical error of these
two standard deviations is about 12%. Hence the internal variances at the two altitudes are





statistically different. This suggests that there may be a vertical structure in the variability that
should be analyzed.
The data sets in Fig. 2 show large changes within short times (2-4 years). Sometimes these
changes are similar at the two altitudes. The variability of HAMMONIA thus appears to
contain an appreciable high frequency component and thus needs to be analyzed as well for
vertical as for spectral structures.
Temperatures at layers 3 km apart in altitude were therefore correlated with those at 42 km
as a reference altitude (near stratopause). The results are shown in Fig.3 for HAMMONIA
model run up to 105 km (red dots). A corresponding  analysis for the much longer model run
of ECHAM6 is also shown ( black squares, up to 78 km). Two important results are obtained:
1) There is an oscillatory vertical structure in the correlation coefficient r with two maxima in
the upper stratosphere and upper mesosphere/lower thermosphere, respectively, and two
minima in the lower stratosphere and in the  mesosphere, respectively (for HAMMONIA).
The correlations are highly significant near the upper three of these extrema (see the 95%
lines in Fig. 3 for HAMMONIA; the significance is much better for ECHAM6). 2.) The
correlations in the two different data sets are  nearly the same above the troposphere. This is
remarkable because the two sets cover time intervals very different in length (34 years vs  400
years, respectively). Therefore, the correlation structure appears to be a basic property of the
atmosphere (see below).

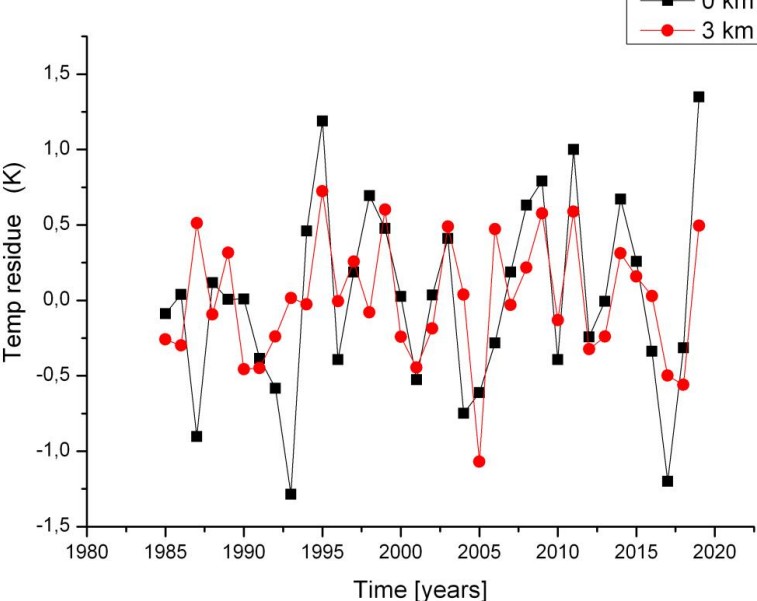

Fig.2   HAMMONIA temperature residues at 0 km and 3 km altitude with fixed boundary
conditions (see text). Mean temperatures of 281.89 K (0 km) and 266.04 K (3 km) have been
subtracted from the model temperatures.
The correlations suggest that the fluctuations in the atmosphere (or part of them) are
somehow "synchronized" at adjacent altitude levels. A vertical (layered) structure might
therefore be present in the magnitude of the fluctuations, too. This was studied by means of
the standard deviations σ of the temperatures T, the result is shown in Fig. 4. There is indeed a
vertical structure with fairly pronounced layers.
The HAMMONIA data used for Fig.4 were annual data that have been smoothed by a four
point running mean. This was done to reduce the influence of high frequency "noise"
mentioned above, which is substantial (a factor of 2).
The layered structures shown in Fig. 3 and 4  are not unrelated. This can be seen in Fig. 4
that also gives  the vertical correlations r (Fig.3) for comparison. The horizontal dashed lines
indicate that the maxima of the standard deviations occur near the extrema of the correlation
profile in the stratosphere and lower mesosphere. This means that the fluctuations in adjacent
σ maxima (and in adjacent layers) are anticorrelated. Surprisingly these anticorrelations are
also approximately seen in the amplitude and phase profiles of Fig.1 that are typical of all
oscillations (see below).

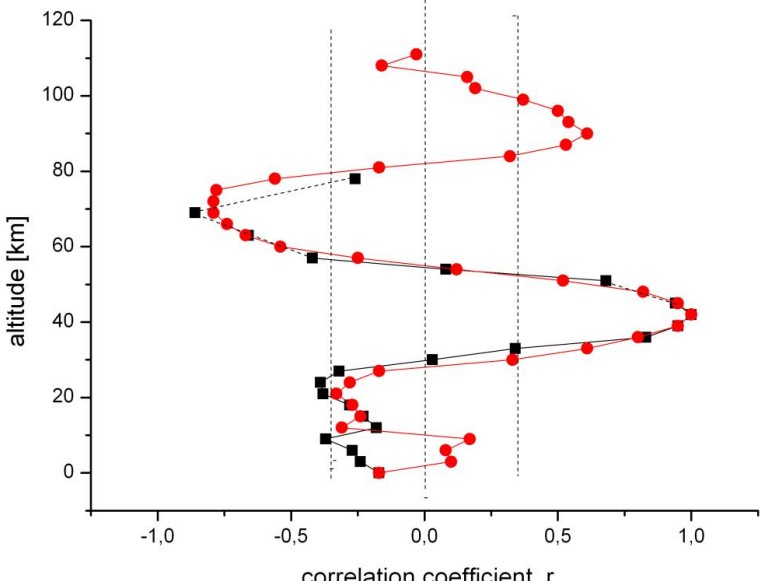

Fig.3   Vertical correlation of temperatures in HAMMONIA (red dots) and ECHAM6 (black
squares). Reference altitude is 42 km (r = 1). Vertical dotted lines show 95% significance for
Hammonia.


The ECHAM6 data have been analyzed in the same way as the HAMMONIA data,
including a smoothing by a 4 point running mean. The data cover the altitude range of 0
-78 km for a 400 year simulation. The results are very similar to those of





HAMMONIA. This is shown in Fig.5 that gives vertical profiles of standard deviations and of
vertical correlations of the smoothed ECHAM6 data, and is to be compared to the
HAMMONIA results in Fig. 4.  The two upper maxima of standard deviations are again
anticorrelated.
It is apparently a basic property of the atmosphere's internal variability to be organized in
some kind of "layers", and that adjacent layers are anti-correlated. It appears therefore
questionable whether the internal variability may be termed "noise", as is frequently done in
the literature.

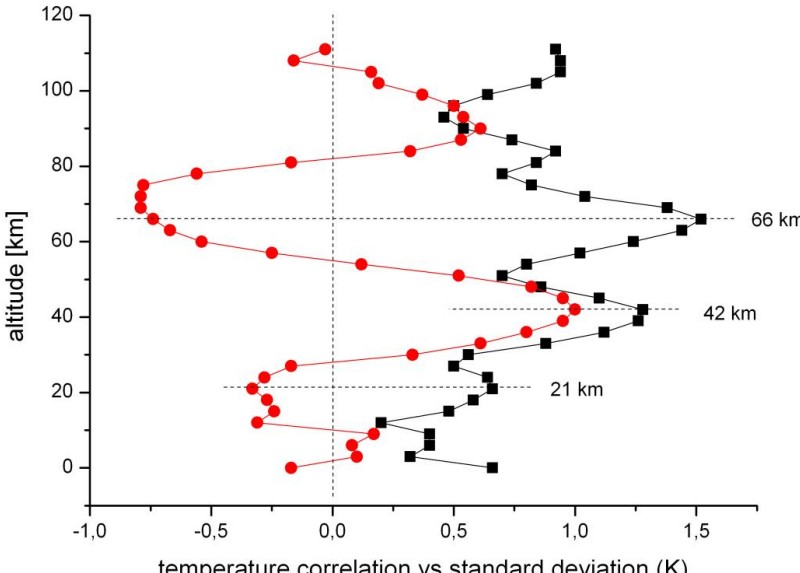

Fig.4   HAMMONIA temperatures:   Comparison of standard deviations (black squares,
multiplied by 2) and correlation coefficients (red dots, see Fig. 3). For details see text.
3.2   Time structures
The correlations/anticorrelations concern temporal variations of temperatures. This suggests
a search for some kind of regular (ordered) structure in the time series, as well. Therefore in a
first step, FFT  analyses have been performed for all HAMMONIA altitude levels (3 km
apart). The results are shown in Fig.6 that gives amplitudes for the period range of 4 - 34
years versus altitude. Also in this picrure, the amplitudes show a layered structur. In addition
an ordered structure in the period domain is also indicated. There are increased or high
amplitudes near certain period values, for instance at the left and right hand side and in the
middle of the picture. A similar result is obtained for the ECHAM6 data shown in Fig.7 for
the longer periods of 10-400 years. The layered structure in altitude is clearly seen, and so are
the increased amplitudes near certain period values. Obviously, the computer simulations
contain periodic temperature oscillations, the amplitudes of which show a vertically layered



Atmospheric
Chemistry
and Physics

Discussions

order. Because the boundary conditions of the computer runs were kept constant, these
oscillations cannot be excited from the outside. They are therefore interpreted as self-excited
(self-sustained) oscillations, and thus as intrinsic properties of the atmosphere

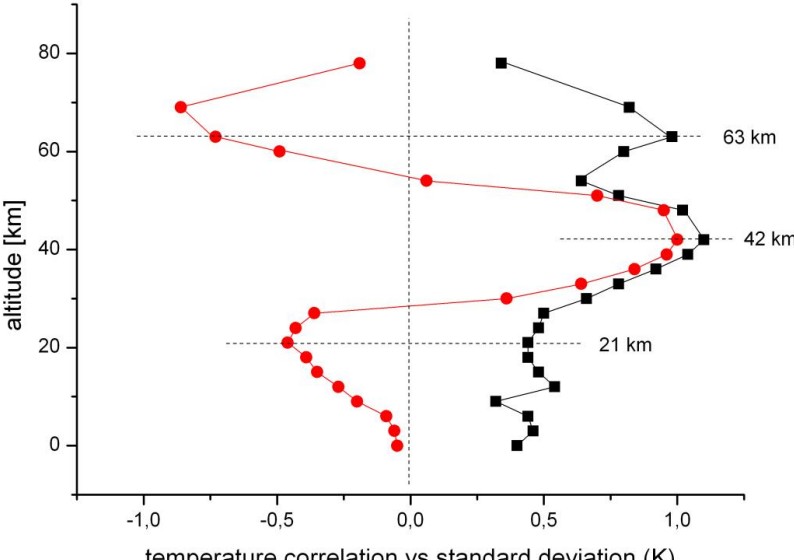

Fig.5   ECHAM6 temperatures:   Comparison of standard deviations (black squares,
multplied by 2) and correlation coefficients (red dots). For details see text.

373       The amplitudes shown in Fig.6 and 7 are relative values, and the resolution of the spectra is
quite limited. Therefore a more detailed analysis is required. For this purpose the Lomb-
Scargle Periodogram (Lomb 1976; Scargle 1982) is used. As an example Fig 8 shows the
mean Lomb-Scargle Periodogram in the period range 20 – 100 years for the ECHAM6 data.
For this picture Lomb-Scargle spectra were calculated for all ECHAM6 layers separately, and
the mean spectrum of all altitudes was determined. The power of the periodogram gives the
reduction in sum of squares when fitting a sinusoid to the data (Scargle 1982), i.e. it is
equivalent to a harmonic analysis using least square fitting of sinusoids. The power values are
normalized by the variance of the data to obtain comparability of the layers with different
variance. Quite a number of spectral peaks are seen between 20 and 60 years period. Further
oscillations appear to be present around 100 years and at even longer periods (not shown here
as they are not sufficiently resolved).
385       We compared the mean result for the ECHAM6 data with 10000 representations of noise.
One representation covers 47 atmospheric layers. For each representation we took noise from
a Gaussian distribution for each atmospheric layer independently, and calculated a mean
Lomb-Scargle Periodogram for every representation in the same way as for the ECHAM6
data. The red line in Fig. 8 shows the average of all of these mean periodograms. As expected
for the average of all representations the peaks cancel, and one gets an approximately constant





value for all periods. A single representation typically shows one or several peaks above this
mean level. The red dashed line gives the upper 2σ level, i.e. the mean plus 2σ. As the mean
Lomb-Scargle Periodogram for the ECHAM6 data shows several peaks clearly above this
upper 2σ level, this mean periodogram is significantly different from that of independent
noise. Therefore, the conclusion is that independent noise at the different atmospheric layers
alone cannot explain the observed periodogram showing large remaining peaks after
averaging. A coupling mechanism between the layers has to be present to explain the
observed mean Lomb-Scargle Periodogram for the ECHAM6 data.
It might be considered appropriate to use red noise instead of white noise in this analysis.
We therefore calculated the sample autocorrelation at a lag of 1 year for the different
ECHAM6 altitudes. These values were found to be very close to zero and, thus, we used
Gaussian noise in our analysis.
The period values shown in Fig. 8 agree with those given for ECHAM6 in Table 2 which
are from the harmonic analysis described next. The agreement is within the error bars given in
Table 2 (except for 24.3).
A spectral analysis as that in Fig.8 was also performed for the HAMMONIA temperatures.
It showed the periods of 5.3 yr and 17.3 yr above the 2σ level. These values agree within
single error bars with those given in Table 2. All peaks found to be significant (in different
analyses) are marked by heavy print in Table 2.

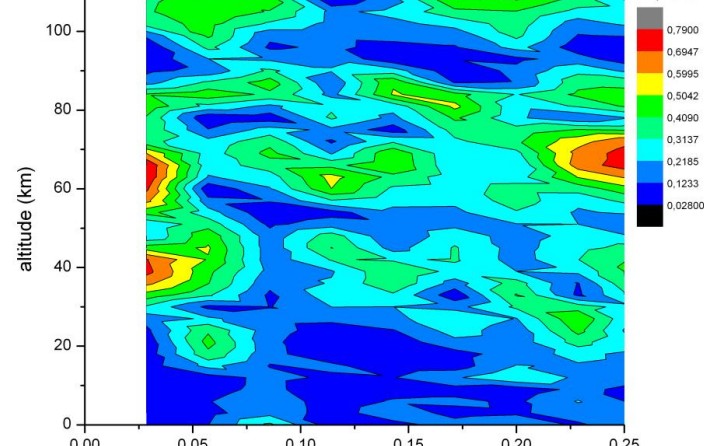

Fig. 6   Self-excited temperature oscillations in the HAMMONIA model.
FFT amplitudes are shown in dependence on altitude and frequency (periods 4 – 34 yr).
Colour code of amplitudes is in arbitrary units.
The Lomb-Scargle spectra (in their original form) do not reveal the phases of the oscillations.
We have therefore added harmonic analyses to our data series. This was done by stepping
through the period domain in steps 10% apart. In each step we looked for the largest near-by
sinus oscillation peak. This was done by means of an ORIGIN search algorithm (ORIGIN Pro
8G, Levenberg-Marquardt algorithm) that yielded optimum values for period, amplitude, and
phase. The results are a first approximation, though, because only one period was fitted at a
time, instead of the whole spectrum. Also, the 10% grid may be sometimes too coarse.



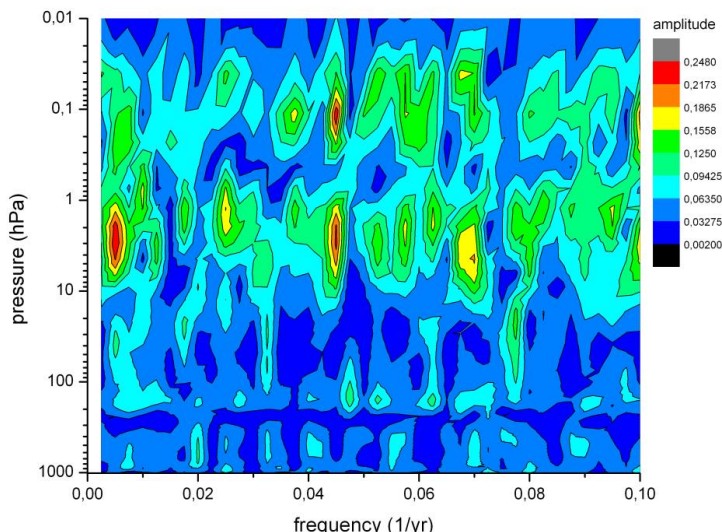

Fig. 7   Self-excited temperature oscillations in the ECHAM6 model.
FFT amplitudes are shown in dependence on altitude and frequency (periods 10 – 400 yr).
Colour code of amplitudes is in arbitrary units.

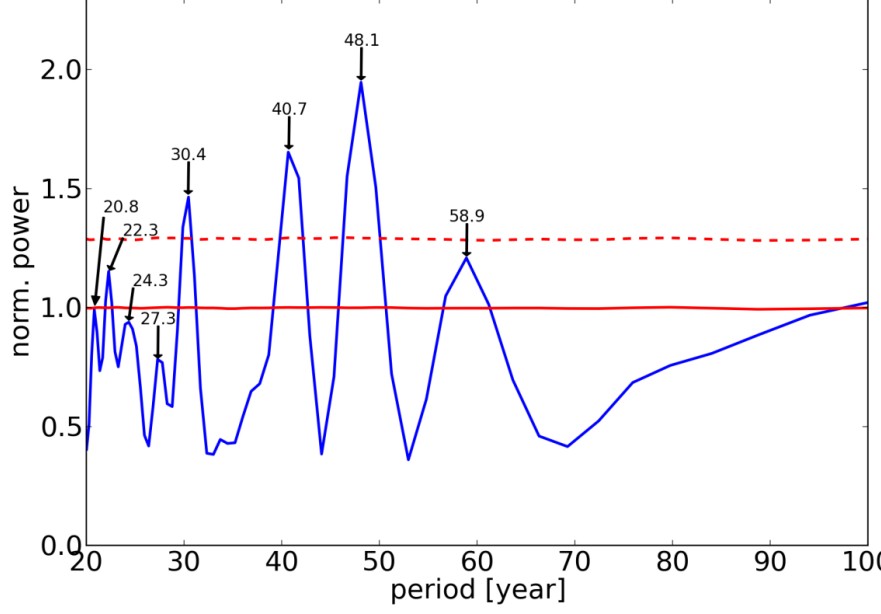

Fig.8   Self-excited temperature oscillations in the ECHAM6 model
Lomb-Scargle periodogram is given for periods of  20 – 100 years. Dashed line indicates
significance at the 2σ level (see text).




438 This analysis was performed for all altitude levels available. Figure 1 shows an example
for the HAMMONIA temperatures from 3-111 km for periods around $15 - 20$ years. The
middle track (red dots) shows the periods with their error bars, the left side shows the
amplitudes, and the right side the phases. The mean of all periods is $17.3 \pm 0.79$ years. There
are several altitudes where the harmonic analysis does not give a period. This may occur if an
amplitude is very small or if there is a  near-by period with a strong amplitude that masks the
smaller one. At these altitudes the periods were interpolated for the fit (dash-dotted vertical
line). The mean of the derived periods (17.3 yr) is used as an estimated interpolation value.
This is because the derived periods do not deviate too much from the mean value. This
procedure allows to obtain estimated amplitude and phase values for instance in the vicinity
of the amplitude minima. That is important because at these altitudes large phase changes are
frequently observed. The harmonic analysis algorithm calculates an amplitude and phase if a
prescribed (estimated) period is provided.
451 The right track in Fig.1 shows the phases of the oscillations. The special feature about this
vertical profile is its steplike structure with almost constant values in some altitudes and a
subsequent fast change somewhat higher to some other constant level. These changes are by
about 180° ($\pi$), i.e. the temperatures above and below these levels are anti-correlated. At these
levels the temperature amplitudes (left track) are minimum, with maxima in between. These
maxima occur near the altitudes of the maxima of the temperature standard deviations in Fig.4
that are anti-correlated in adjacent layers. The phase steps in Fig.1 approximately fit to this
picture. They suggest that the layer anti-correlation discussed above is at least in part due to
the phase structure of the self-sustained oscillations in the atmosphere.
460 This important result was checked by an analysis of other oscillations contained in the
HAMMONIA data series. Nine self-sustained oscillations with periods between 5.34 years
and 28.5 years were obtained by the analysis procedure described above. They are listed in
Table 2, and all show vertical profiles similarly as in Fig.1.
464 Figure 1 shows that at different altitudes the periods are somewhat different. They cluster,
however, quite closely about their mean value of 17.3 yr. This clustering about a mean value
is found for all periods listed in Table 2. This is shown in detail in Fig. 9 and 10 which give
the number of periods found at different altitudes in a fixed period interval. The clusters are
separated by major gaps, as is indicated by vertical dashed lines (black). This suggests to use
a mean period value as an estimate of the oscillation period representative for all altitudes.
The mean period values are given above each cluster in red, together with a red solid vertical
line. A few clusters are not very pronounced, and hence the corresponding mean period values
are unreliable (e.g. 22.8 yr, see the increased standard deviations in Table 2).
473 ECHAM6 -  data are used in the present paper to analyze much longer time windows (400
years) than that of HAMMONIA (34 years). Results shown in Fig. 3, 5, and 7 are quite
similar to those of HAMMONIA. Harmonic analysis of self-sustained oscillation periods was
performed in the same way as for HAMMONIA. Eighteen periods were found longer than 20
years and have been included to Table 2. Shorter periods are not shown here as that range
is covered by HAMMONIA. The amplitude and phase structures of these are very similar to

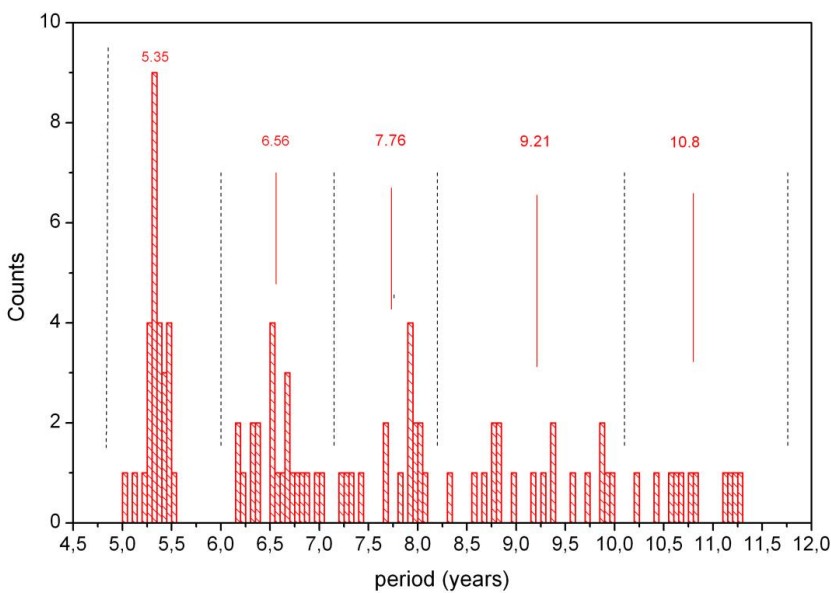

Fig.9   Number of oscillations counted in a fixed period interval at periods 4.75 – 11.75 years.
Interval is 0.05 years. (HAMMONIA)

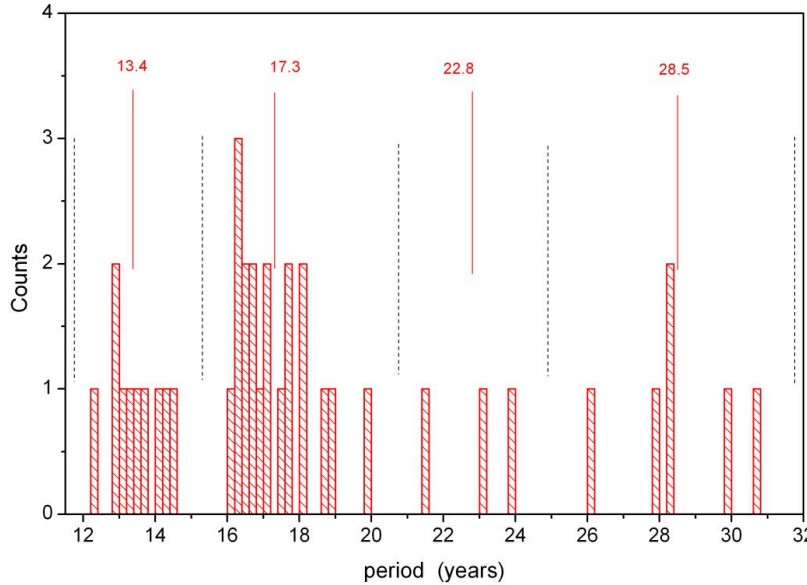

Fig.10   Number of oscillations counted in a fixed period interval at periods 11.75 – 31.75
years. Interval is 0.2 years. (HAMMONIA)





those of HAMMONIA. The cluster formation about the mean period values is also obtained
for ECHAM6 and looks quite similar to Fig.9 and 10.
The vertical amplitude and phase profiles of the mean periods given in Table 2 all show
intermittend amplitude maxima/minima, and step-like phase structures. They in general look
very similar to Fig.1. We have calculated the accumulated amplitudes (sums) from all of these
profiles at all altitudes. They are shown in Fig.11 (for HAMMONIA). They clearly show a
layered structure similar to the temperature standard deviations in Fig 4, with maxima at
altitudes close to those of the standard deviation maxima. The figure also closely corresponds
to the amplitude distribution shown in Fig.1, with maxima and minima occurring at similar
altitudes in either picture.

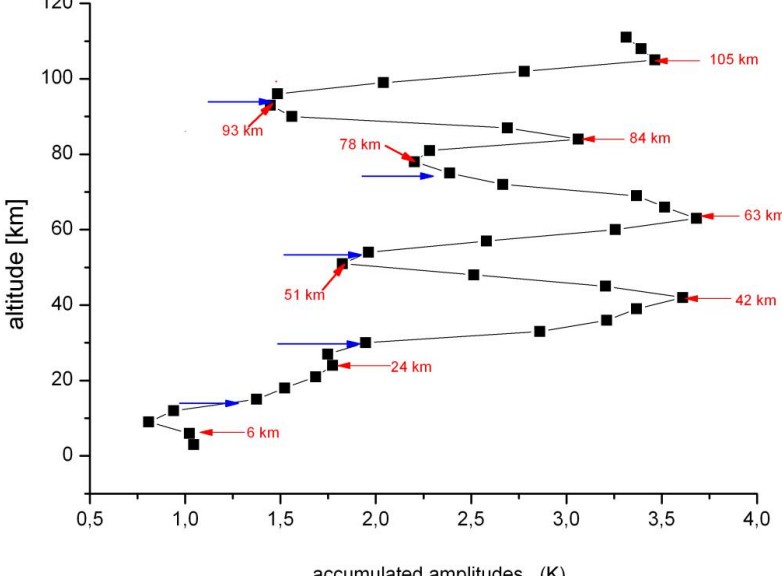

Fig. 11   Self-excited temperature oscillations in the HAMMONIA model.
Accumulated amplitudes are shown vs altitude for periods of 5.3 – 28.5 years (see Table 2).
Blue horizontal arrows show mean altitudes of phase jumps. Red arrows indicate altitudes of
maxima and minima.
Accumulated amplitudes have also been calculated for the ECHAM6 periods, and very
similar results are obtained as for HAMMONIA. The similarity is already indicated in Fig.3
above 15 km. The correlation of the HAMMONIA and ECHAM6 curves above this altitude
has a correlation coefficient of 0.97. This is remarkable because many more oscillations are
contained in the ECHAM6 data set than in HAMMONIA, as it was mentioned in Sect. 3.1.
This and Fig.11 supports the idea that all self-excited oscillations have about the same vertical
amplitude structure.
The phase jumps in the nine oscillation vertical profiles of HAMMONIA also occur at
similar altitudes. Therefore the mean altitudes of these jumps have been calculated and are





shown in Fig.11 as blue horizontal arrows. They are seen to be close to the minima of the
accumulated amplitudes and thus confirm the anticorrelations between adjacent layers.
Figures 4, 1, and 11 thus show a general structure of temperature correlations/anticorrelations
between different layers of the HAMMONIA atmosphere, and suggest the   phase structure of
the self-sustained oscillations as an explanation. The same is valid for ECHAM6.
Altogether HAMMONIA and ECHAM6 consistently show the same type of variability and
oscillation structures. This type occurs in a wide time domain of 400 years. As mentioned, we
do not believe that these ordered structures are adequately described by the term "noise", as
this notion is normally used for something occurring at random.

3.3     Intrinsic oscillation periods

Three different model runs of different lengths have been investigated by the harmonic
analysis described. The HAMMONIA model covered 34 years, the WACCM model covered
150 years, and the ECHAM6 model covered 400 years. The intention was to study the
differences resulting from the different nature of the models, and from the difference in the
length of the model runs.
The oscillation periods found in these model runs are listed in Table 2. These periods are
vertical mean values as described for Fig.1 and Figs. 9-10. Periods are given in order of
increasing values in years together with their standard deviations. Only periods longer than 5
years are shown here. The maximum period cannot be longer than the length of the computer
run. Therefore, the number of periods to be found in a model run can -in principle-  be the
larger the longer the length of the run is. Table 2 shows preferentially periods longer than 20
yr (except for HAMMONIA and Hohenpeißenberg) as the emphasis  is on the long periods
here.
Table 2 also contains two rows of periods and their standard deviations that were derived
from *measured* temperatures. These are data obtained on the ground at the Hohenpeißenberg
Observatory (47.8°N, 11.0°E) from 1783 to 1980, and globally averaged GLOTI data (Global
Land Ocean Temperature Index , Hansen et al., 2010), respectively. The data are annual mean
values smoothed by a 16 point running mean and will be discussed below. Data after 1980 are
not included in the harmonic analyses because they steeply increase thereafter ("climate
change"). The periods are determined as for the zero level data of the other rows of Table 2
(see Section 3.2).
There are some empty spaces in the lists of Table 2. It is believed that this is because these
oscillations are not excited in that model run, or that their excitation is not strong enough to be
detected, or that the spectral resolution of the data series is insufficient (strong changes in
amplitudes strengths are, for instance, seen in Fig. 1.). For the *measured* data in Table 2 it
needs to be kept in mind that they were under the influence of varying boundary conditions.
The model runs shown in Table 2 have different altitude resolutions. The best resolution (1
km) is available in HAMMONIA (119 vertical layers, run Hhi-max in the earlier paper of
Offermann et al., 2015). The very long run of ECHAM6 uses only 47 layers. Data on a 3 km
altitude grid are used here. In the earlier paper it was shown on the basis of a limited data set
(HAMMONIA, Hlo-max) that a decrease of the number of layers affected the vertical
amplitude and phase profiles of the oscillations found. It did, however, not change the
oscillation periods. For a more detailed analysis a 20 year-long run of Hlo-max (67 layers) is
now compared to the 34 year- long run of Hhi-max (119 layers). The resulting oscillation
periods are shown in Table 3 (together with their standard deviations). Sixteen pairs of
periods are listed that all agree within the single error bars (except No. 4). Hence it is
confirmed that the periods of the oscillations are quite robust with respect to changes in



altitude resolution. The periods of the ECHAM6 run can therefore be considered as reliable,
despite their limited altitude resolution.

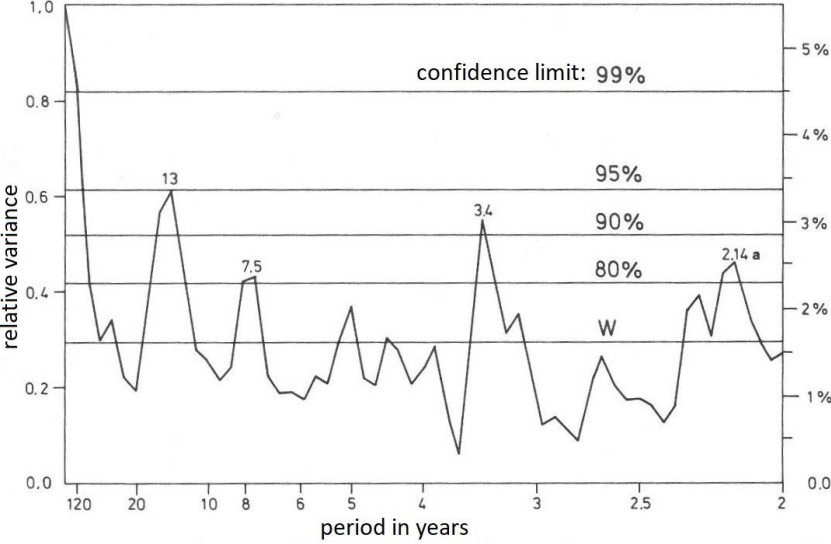

Fig.12    Periodogram (2 yr to 20 yr) of measured Hohenpeißenberg temperatures from
Schönwiese (1992, Abb.57). Results are from an autocorrelation spectral analysis ASA.
When comparing the periods in Table 2 to each other several surprising agreements are
observed. It turns out that all periods of the HAMMONIA and WACCM models find a
counterpart in the ECHAM6 data (not vice versa). These data pairs always agree within their
combined error bars, and mostly even within single error bars. The difference between the
members of a pair is much smaller than the distance to any neighbouring value with higher or
lower ordering number in Table 2. From this it is concluded that the different models find the
same oscillations. The periods of them are obviously quite robust. This and the fact that the
boundary conditions have been kept constant makes us believe that these oscillations are self-
sustained (intrinsic) oscillations.
A similar agreement is seen for the periods found in the measured Hohenpeißenberg data,
although these have been under the influence of variations of the sun, ocean, and greenhouse
gases. A spectral analysis (auto correlation spectral analysis ASA) of these data is shown in
Fig.12. It was taken from Schönwiese (1992). The important peak at 3.4 years is not
contained in Table 2, but was found in Offermann et al. (2015). The two peaks near 7.5 yr and
13 yr are close to the values 7.83 ± 0.26 yr and 13.6 ± 0.8 yr in Table 2.
A 335 year long data set of Central England Temperatures (CET) is the longest measured
temperature series available (Plaut et al., 1995). A singular spectrum analysis was applied by
these authors for interannual and interdecadal periods. Periods of 25.0 yr, 14.2 yr, 7.7 yr, and
5.2 yr were identified. All of these values nearly agree with numbers given for HAMMONIA,
WACCM, and/or ECHAM6 in Table 2 (within the error bars given in the Table).
Meyer and Kantz (2019) recently studied the data from a large number of European stations
by the method of detrended fluctation analysis. They identified a period of 7.6 ± 1.8 yr, which
again is in agreement with the HAMMONIA results given in Table 2 (and also agrees with
Fig.12, and with Plaut et al.,1995).



Also the GLOTI data in Table 2 are in agreement with some of the other periods, even
though they are global averages. The results altogether suggest that the periods discussed are
basic (intrinsic) properties of the atmosphere. It will be shown below that they are not limited
to atmospheric temperatures alone, but are, for instance, also seen in Methane mixing ratios.

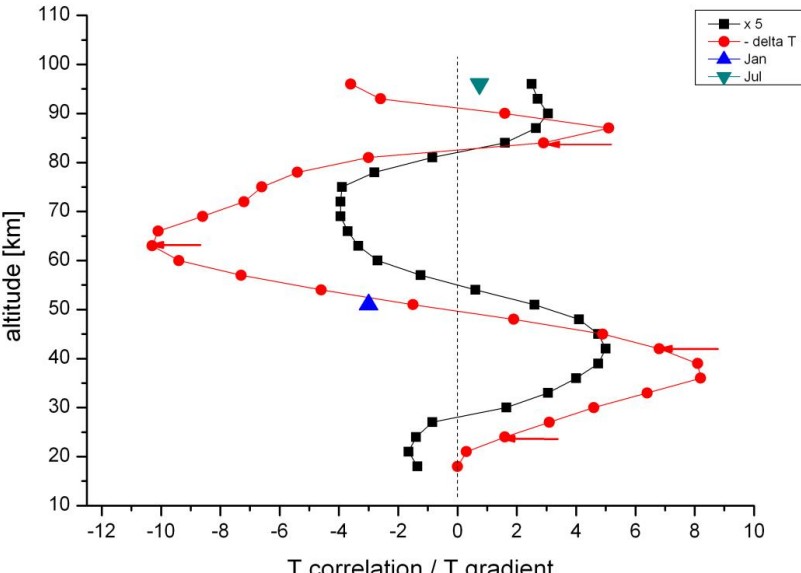

Fig.13     Comparison of  HAMMONIA vertical correlations from Fig.3 (black squares) with
vertical temperature gradients (red dots). Data are from annual mean temperatures.
Correlation coefficients are multiplied by 5. Temperature gradients are approximated by the
differences of consecutive temperatures (K per 3 km). Two gradients are given for monthly
mean temperature curves in addition: blue triangle for January, green inverted triangle for
July. Red arrows show the altitudes of the maxima of the accumulated amplitudes in Fig.11.
613         3.4   Oscillation amplitudes
In an attempt to learn more about the nature of the self-sustained oscillations we analyze
their oscillation amplitudes. The determination of absolute amplitudes of self-excited
oscillations is difficult and beyond the scope of the present paper. Nevertheless, interesting
results can be obtained from their relative values. One of these results is related to the vertical
gradients of the atmospheric temperature profiles.
The HAMMONIA model simulates the atmospheric structure as a whole. The annual mean
vertical profile of HAMMONIA temperatures can be derived and is seen to vary between a
minimum at the tropopause, a maximum at the stratopause, and another minimum near the
mesopause (not shown here). In consequence the vertical temperature gradients change from
positive to negative, and to positive again. This is shown in Fig.13 (red dots) between 18 km
and 96 km. The temperature gradients are approximated by the temperature differences of
consecutive levels.


Also shown in Fig.13 is the correlation profile of HAMMONIA from Fig.3 (black squares
here). The two curves are surprisingly similar (correlation coefficient is 0.80. Outside the
range shown the correspondence is lost.). The similarity suggests some connection of the
oscillation structure and the mean thermal structure of the middle atmosphere. This is
supported by the accumulated amplitudes of the self-excited oscillations in Fig.11. The
maxima of these occur at altitudes near to the extrema of the temperature gradients as is
shown by the red arrows in Fig. 13. The mechanism connecting the oscillations and the
thermal structure appears to be active throughout the whole altitude range shown (except the
lowest altitudes).
A possible mechanism might be a vertical displacement of air parcels. If an air parcel is
displaced vertically by some distance D ("displacement height")  a relative change in mixing
ratio is observed that can be estimated by the product {D times mixing ratio gradient}. If the
vertical movement is an oscillation the trace gas variation is an oscillation as well, assuming
that D is a constant. Such transports may be best studied by means of a trace gas like CH4.
HAMMONIA methane mixing ratios have therefore been investigated for oscillation periods
in the same way as described above for the temperatures. Results are briefly summarized here.
Ten periods have been found, indeed, between 3.56 and 16.75 years by harmonic analyses.
These periods are very similar to those obtained for the temperatures in Table 2. The
agreement is within the single error bars. Hence it is concluded that the same self-sustained
oscillations are seen in HAMMONIA temperatures and CH4 mixing ratios.
The CH4 oscillations support the idea that a displacement mechanism is active. The
corresponding displacement heights D were estimated from the CH4 amplitudes and the
vertical gradients of the mean HAMMONIA CH4 mixing ratios.
The values D obtained from the different oscillation periods are about the same, though they
show some scatter. This means that the displacement mechanism is the same for all
oscillations. However, D appears to follow a trend in the vertical direction. The displacements
are below 100 m in the lower stratosphere and slowly increase with height to above 200 m.
Thus the important result is obtained that the self-sustained oscillations are related to a
vertical displacement mechanism that is altitude dependent, but appears to be the same for  all
periods. A more detailed analysis is beyond the scope of this paper.
3.5  Seasonal aspects
Our analysis has so far been restricted to annual mean values. Large temperature variations
on much shorter time scales are also known to occur in the atmosphere, including vertical
correlations (e.g. seasonal variations). This suggests the question whether these might be
somehow related to the self-excited, long period oscillations. Our spectral analysis is therefore
repeated using monthly mean temperatures of HAMMONIA.
Results are shown in Fig. 14 and 15, which give the amplitude distribution vs period and
altitude of FFT analyses for the months of July and January. These two months are typical of
summer (May-August), and winter (November-March), respectively. In July oscillation
amplitudes are seen essentially at altitudes above about 80 km, and some below about 20 km.
In the regime in between, oscillations are obviously very small or not excited. The opposite
behaviour is seen in January: oscillation amplitudes are now observed in the middle altitude
regime where they had been absent in July. This is to be compared to Fig.6 and 11 that give
the annual mean picture. In Fig. 11 the structures (two peaks) above 80 km appear to
represent the summer months (Fig. 14). The structures between 80 km and 30 km, on the
other hand,  apparently are representative of the winter months (Fig. 15).
The monthly oscillations appear to be linked to the wind field of the HAMMONIA model.
Figure 16 shows the monthly zonal winds of HAMMONIA from the ground up to 111 km





(50°N). Comparison with Fig. 14 and 15 shows that oscillation amplitudes are obviously not
observed in an easterly wind regime. Hence, the long period self-sustained oscillations and
their phase changes are apparently linked to the dynamical structure of the middle
atmosphere. A change from high to low oscillation activity in the vertical direction appears to
be linked to a wind reversal.
This correspondence does not, however, exist in all details. In the regimes of oscillation
activity there are substructures. For instance in the middle of the July regime of amplitudes
above 80 km there is a "valley" of low values at about 95 km. A similar valley is seen in the

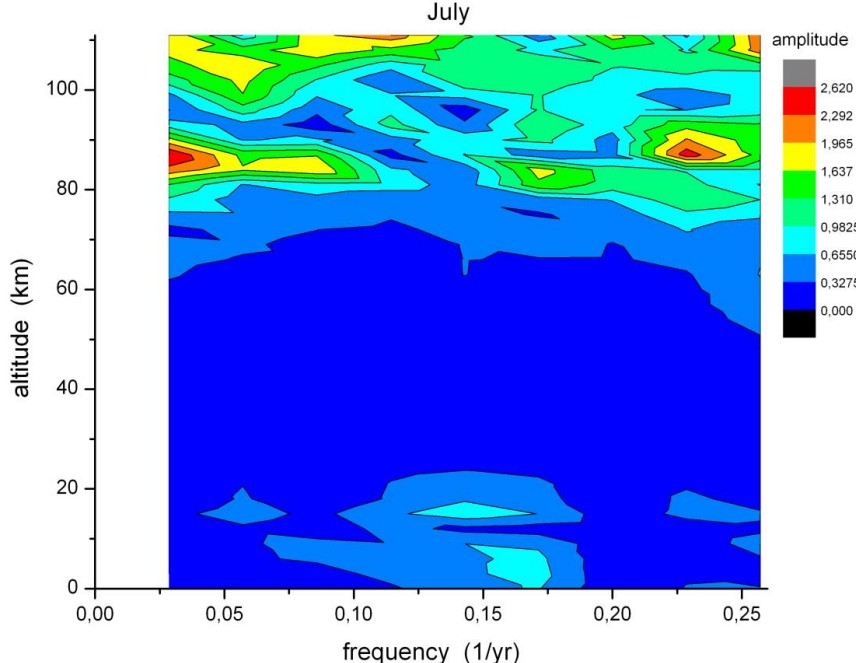

Fig. 14 Self-excited temperature oscillations in the month of July in HAMMONIA.
Amplitudes are shown in dependence of altitude and frequency (periods 3.9-34 yr). Colour
code of amplitudes is in arbitrary units.
January data around 55 km. Near these altitudes there are phase changes of about 180° (see
the blue arrows in Fig.11). Contrary to our expectation sketched above, these are altitudes of
large westerly zonal wind speeds without much vertical change (see Fig.16). However, the
two "valleys" are relatively close to altitudes where the vertical temperature gradients are
small (see Fig.13). As the gradients from the annual mean temperatures used for the curves in
Fig.13 may differ somewhat from the corresponding monthly values two monthly gradients
have been added in Fig.13 for January (at 51 km) and at 96 km (for July). They are small,
indeed, and could explain low oscillation amplitudes by the above discussed vertical
displacement mechanism.
703        3.6   Oscillation persistence
If our concept of self- excitation of oscillations is correct we might expect that such
oscillations might also dissipate after a while, i.e. we should expect some intermittance in our



oscillation amplitudes. To check on this we have subdivided the 400 years data record of
ECHAM6 in four smaller time intervals (blocks) of 100 years each. In each block we
performed harmonic analyses for periods of 24 yr (frequency 0.042/yr) and 37 yr (frequency
0.027/yr), respectively, at the altitudes of 42 km (1.9 hPa) and 63 km (0.11 hPa), respectively.
These are altitudes and periods with strong signals as seen in Fig.7. Results for the two
altitudes and two periods are given in Fig.17.
The results show two groups of amplitudes: one is around 0.15 K, the other is very small
and compatible with zero. The two groups are significantly different as is seen from the error
bars. This result is compatible with the picture of oscillations being excited and not-excited
(dissipated) at different times. The non-excitation (dissipation) for the 24 yr oscillation (black
squares) occurs in the first block (century), that for the 37 yr oscillation (red dots) in the
second block. The 24 yr profile at 63 km altitude is similar as that at 24 km. Likewise, the 37
yr profile at 24 km is similar to that at 63 km. Hence it appears that the whole atmosphere (or
a large part of it) is excited (or dissipated) simultaneously. (The two profiles in Fig.17 appear
to be somehow anticorrelated for some reason that is unknown as yet.)

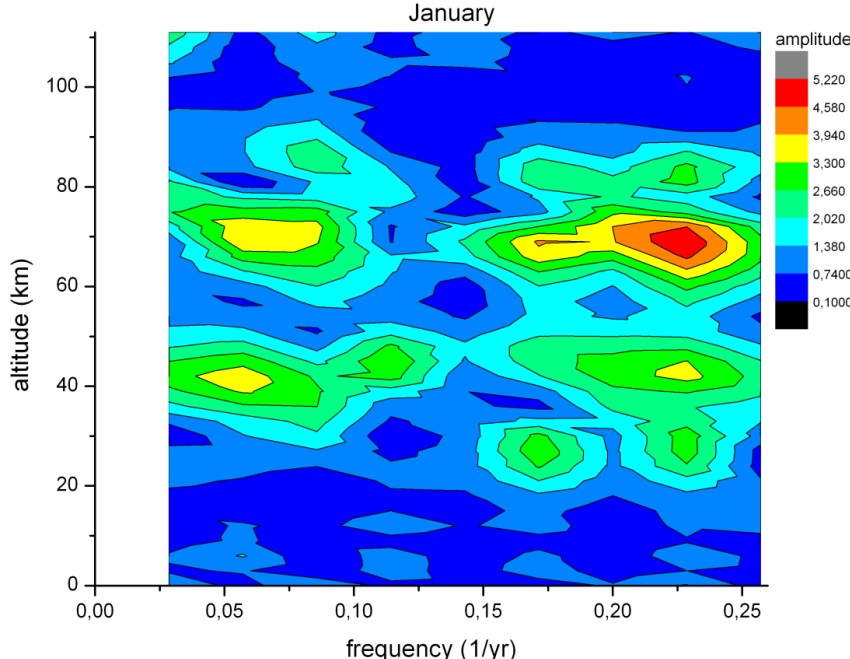

Fig. 15  Self-excited temperature oscillations as in Fig. 14, but for the month of January

For the analysis of shorter periods the 400 year data set of ECHAM6 may be subdivided in
a larger number of time intervals. Figure 18 shows the results for periods of 5.4 yr and 16 yr,
respectively, for various altitudes. An FFT analysis was performed in 12 equal time intervals
(blocks of 32 yr length) in the altitude regime 0.01 – 1000 hPa and the period regime 2 – 40
yr. The corresponding 12 maps look similar as Fig.15, i.e. there are pronounced amplitude hot
spots at various altitudes and periods. In subsequent blocks these hot spots may shift
somewhat in altitude and/or period, and hence the profiles taken at a fixed period and altitude
as those of Fig.18 show some scatter. Nevertheless, there is strong indication of the
occurrence of coordinated high maxima and deep minima of amplitudes in Blocks 3/ 4 and





Blocks 10/11, respectively. These maxima are interpreted as strong oscillation excitation,
whereas the minima are believed to show (at least in part) the dissipation of the oscillations.
It should be mentioned that in the FFT analysis the 5.4 yr period is an overtone of the 16 yr
period. Hence the two period data in Fig.18 may be somehow related.

Fig.16  Vertical distribution of zonal wind speed in the HAMMONIA model.
4   Discussion
4.1  The nature and origin of the self-sustained oscillations are as yet unknown. We
therefore collect here as many of their properties as possible. They do exist in computer
models even if the model boundaries for the influences of the sun, the ocean, and the green
house gases are kept constant. Therefore they are believed to be self-generated oscillations.
Further properties are as follows: The periods are robust, i.e. they are found with similar
values in different models. The periods cover a wide range from 2 to 341 years (at least).  The
different oscillations have similar vertical profiles (up to 110 km) of amplitudes and phases.
This may indicate three-dimensional atmospheric oscillation modes. To clarify this, latitudinal
and longitudinal studies of the oscillations are needed in a future analysis.
4.2  The accumulated oscillation amplitudes show a layer structure with alternating maxima
and minima and correlations / anticorrelations in the vertical direction. These appear to be
influenced by the seasonal variations of temperature and zonal wind in the stratosphere,
mesosphere, and lower thermosphere. Table 4 summarizes the results shown in Section 3.5.
Maxima of oscillation amplitudes appear to be associated with westerly (eastward) winds
together with large temperature gradients (positive or negative). Amplitude minima are
associated with either easterly (westward) winds or with near zero temperature gradients. The
latter feature is compatible with a possible vertical displacement mechanism. Such
displacements can be seen, indeed, in the CH4 data of the HAMMONIA model.  The
mechanism summarized in Table 4  appears to be a

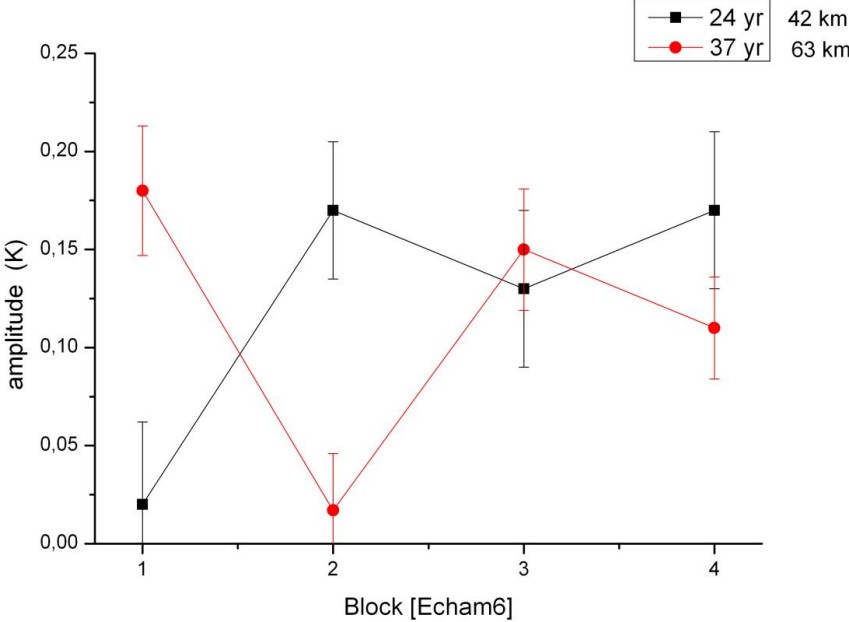

Fig.17   Amplitudes of 24 yr and 37 yr oscillations in four subsequent equal time intervals
(Blocks) of the 400 year data set of ECHAM6.

basic feature of the atmosphere that influences many different parameters as temperature,
mixing ratios, etc.. Vertical displacements of measured temperature profiles have been
discussed for instance by Kalicinsky et al.(2018).

4.3   The amplitudes found for the self-sustained oscillations are relatively small (Fig.1). The
question therefore arises whether these oscillations might be spurious peaks, i.e. some sort of
noise. We tend to deny the question for the following reasons:

(a)   An accidental agreement of periods as close together as those shown in Table 2 for
different model computations appears very unlikely. This also applies to the Hohenpeißenberg
data in Table 2, and several of these periods are even found in the GLOTI data.
788       If the period values were accidental they should be evenly distributed over the
period- space. To study this the range of ECHAM6 periods (20 – 341 yr) is
considered. Table 2 shows that the error bars (standard deviations) of ECHAM6
cover approximately half of this range. If the periods of this and some other data set occur at
random, half of them should coincide with the ECHAM6 periods within the





ECHAM6 error bars, and half of them should not. This is checked by means of the
WACCM model data, the Hohenpeissenberg measured data, and three further
measurements sets that reach back to 1783 (Innsbruck, 47.3°N;11.4°E; Vienna,
48.3°N;16.4°E; Stockholm, 59.4°N;18.1°E). The result is that about two thirds of the
periods coincide with ECHAM6 periods within the ECHAM6 error bars. This is far
from an even distribution.
It is important to note that the data sets used here are quite different in nature: They are
either model simulations with fixed or partially fixed boundaries, or they are real atmospheric
measurements at different locations.

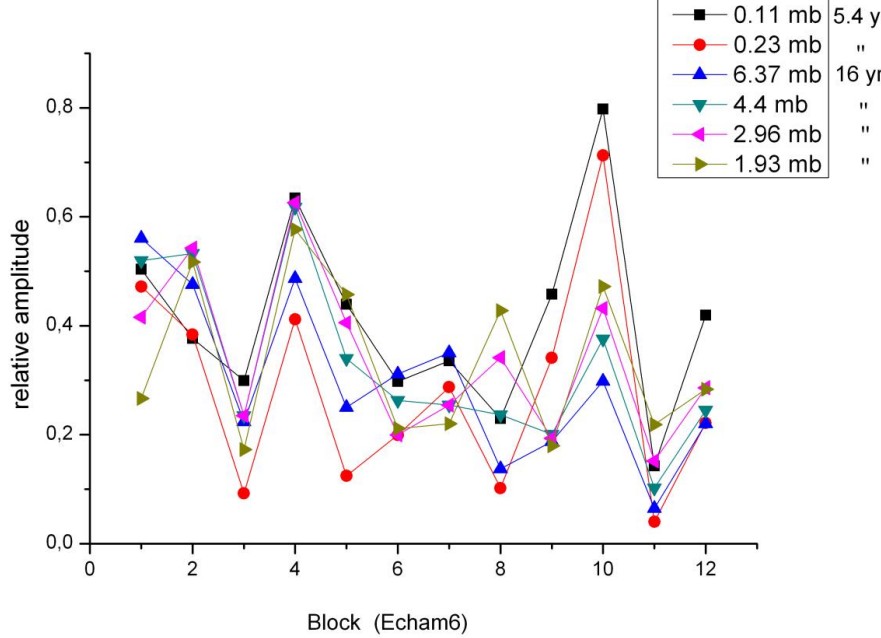

Fig.18   FFT amplitudes of 5.4 yr and 16 yr oscillations in 12 equal time intervals (32 yr
blocks) of the ECHAM6 400 year data set.
A further argument against noise is the distribution of the data in Fig. 9 and 10. If our
oscillations were noise, the peaks in these Figures should be evenly distributed with respect to
the period scale. However, the distribution is highly uneven, with high peaks and large gaps,
which is very unlikely to result from noise.
(b)   The periods given in Table 2 were all calculated by means of  harmonic analyses. This
was done to support the reliability of the comparison of the three models and four
measured data sets. There could be, however, the risk of a "common mode failure".  The
harmonic analysis results are therefore checked, and are confirmed by the Lomb-
Scargle and autocorrelative spectral (ASA) analyses shown in Fig.8 and 12, and by the above
cited results of Plaut et al.,(1995) and Meyer and Kantz (2019). There is, however, not a one-
to-one correspondence of these numbers and those of Table2. In general the number of



oscillations found by the harmonic analysis is larger. Hence several of the Table 2 periods
might be considered questionable. It is also not certain that Table 2 is exhaustive.
Nevertheless, the large number of close coincidences is surprising.
(c)  The layered structure of the occurrence of the oscillations (e.g. Fig.11) and the
corresponding anti-correlations appear impossible to reconcile with a noise field. These
correlations extend over about 20 km (or more) in the vertical which is about three scale
heights. Turbulent correlation would, however, be expected over one transport length, i.e. one
scale height, only.
(d)  The apparent link of the oscillations to the zonal wind field and the vertical
temperature structure (Table 4) would be very difficult to be explained by noise.
(e) The close agreement (within single error bars) of the oscillation periods in
temperatures and in CH4 mixing ratios would also be very difficult to be explained by
noise.
In  summary  it  appears  that  many  of  the  oscillations  are  intrinsic  properties  of  the
atmosphere that are also found in sophisticated simulations of the atmosphere.
4.4   The self-sustained oscillations are studied here mainly for atmospheric temperatures.
They show up, however, in a similar way  in other parameters as winds, pressure, trace gas
densities, NAO, etc.  (Offermann et al., 2015). Some of the periods in Table 2 appear to be
similar to the internal decadal variability of the atmosphere/ocean system (e.g., Meehl et al.,
2013; 2016; Fyfe et al. 2016). One example is the Atlantic Multidecadal Oscillation (AMO)
as discussed by Deser et al.(2010) with time scales of 65-80 yr, and with its "precise nature
…still being refined".  Variability on centennial time scales and its internal forcing was
recently discussed by Dijkstra and von der Heydt (2017). It needs to be emphasized that the
oscillations discussed in the present paper are not influenced by the ocean as they occur even
if the ocean boundaries are kept constant.
4.5   The self-sustained oscillations obviously are somehow related  to the "internal
variability" discussed in the atmosphere/ocean literature at 40 – 80 years time scales ("climate
noise", see e.g. Deser et al., 2012, Gray et al., 2004, and other references in Section 1). The
particular result of  the present analysis is its extent from the ground up to 110 km, showing
systematic structures in all of this altitude regime. These vertical structures lead us to hope
that the nature of the oscillations and hence of (part of) the "internal variability" can be
revealed in the future.
4.6  It appears that the time persistency of the self-sustained oscillations is limited. Longer
data sets are needed to study this further.
4.7   The internal variability in the atmosphere/ocean system  "…makes an appreciable
contribution to the total… uncertainty in the future (simulated) climate response…" (Deser et
al., 2012). Similarly our self-sustained oscillations might interfere with long term (trend)
analyses of various atmospheric parameters. This includes slow temperature increases as part
of the long term climate change, and needs to be studied further.



## 5 Summary and Conclusions

The structures analyzed in this paper are believed to be oscillations that are self-generated (self-sustained) in the atmosphere. The oscillations occur in a similar way in different atmospheric climate models, and even if the boundary conditions of sun, ocean, and greenhouse gases are kept constant. They also occur in long-term temperature measurements series. They are characterized by a large range of period values from below 5 to beyond 300 years. Periods of self-excited oscillations are known to be robust. This is in line with the fact that we find very nearly the same periods in different climate model calculations as well as in long observation series.

As we do not yet understand the nature of the oscillation structures we try to assemble as many of their properties as possible. The oscillations show typical and consistent structures in their vertical profiles. Temperature amplitudes show a layered behaviour in the vertical direction with alternating maxima and minima. Phase profiles are also layered with 180° phase jumps near the altitudes of the amplitude minima (anticorrelations). There are also indications of vertical transports suggesting a displacement mechanism in the atmosphere. As an important result we find that for all oscillation periods the altitude profiles of amplitudes and phases as well as the displacement heights are nearly the same. This leads us to suspect an atmospheric oscillation mode.

These signatures are found to be linked to the thermal and dynamical structure of the middle atmosphere. They are seen to be an essential part of atmospheric dynamics. All results presently available are local, i.e. they refer to the latitude and longitude of Central Europe. In a future step horizontal investigations need to be performed to check on a possible modal structure.

Most of the present results are for temperatures at various altitudes (up to 110 km). Other atmospheric parameters indicate a similar behaviour and need to be analyzed in detail in the future. Also, the potential of the long period oscillations to interfere with trend analyses needs to be investigated.





Author contribution
DO performed data analysis and prepared the manuscript and figures with contributions from
all co-authors.
JW managed data collection and performed FFT spectral analyses.
ChK performed Lomb-Scargle spectral and statistical analyses
RK provided interpretation and editing of the manuscript, figures, and references.
Competing Interests
The authors declare that they have no conflict of interest.



Acknowledgements

Global Land Ocean Temperature Index (GLOTI) data were downloaded from
http://data.giss.nasa.gov/gistemp/tabledata_v3/GLB.Ts+dSST.txt and are gratefully
acknowledged..

We thank Katja Matthes (GEOMAR, Kiel, Germany) for making available the WACCM4
data, and for helpful discussions. Model integrations of the CESM-WACCM Model have
been performed at the Deutsches Klimarechenzentrum (DKRZ) Hamburg, Germany. The help
of Sebastian Wahl in preparing the CESM-WACCM data is greatly appreciated.

HAMMONIA and ECHAM6 simulations were performed at and supported by the German
Climate Computing Centre (DKRZ). Many and helpful discussions with Hauke Schmidt (MPI
Meteorology, Hamburg, Germany) are gratefully acknowleged.

We are grateful to Wolfgang Steinbrecht (DWD, Hohenpeißenberg Observatory, Germany)
for the Hohenpeißenberg data and many helpful discussions.
Part of this work was funded within the project MALODY of the ROMIC program of the
German Ministry of Education and Research under Grant No. 01LG1207A



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





Table 1
Properties of the GCM simulations
All data are for Central Europe (50°N, 7°E), for details see text.

|  | HAMMONIA | WACCM4 | ECHAM6 |
|---|---|---|---|
| Horizontal resolution | T31 | 1.9°x2.5° (lat/long) | T63 |
| Vertical resolution | 119 levels 1 km (stratosphere) | 66 levels | 47 levels |
| altitude range | 0 – 110 km | 0 – 108 km | 0 – 78 km |
| length of simulation | 34 yr | 150 yr | 400 yr |
| time resolution of data used | annual/monthly | annual | annual |
| boundary conditions |  |  |  |
| - sun | fixed | variable (see text) | fixed |
| - ocean | SST fixed | climatological SST and sea ice | fixed |
| - greenhouse gases | fixed | fixed (1960 values) | fixed |
| References | Schmidt et al., 2010 | Hansen et al., 2014 | Stevens et al., 2013 |




Table 2:
Periods of  temperature oscillations from harmonic analyses
Periods are numbered according to increasing values. Periods (in years) are given with their standard deviations.
Self-sustained periods are from the HAMMONIA, WACCM, and  ECHAM6  models, respectively. Additional
periods are from Hohenpeißenberg measurements, and from the Global Land Ocean Temperature Index
(GLOTI).
HAMMONIA periods are limited to 28.5 yr as the model run covered 34 yr , only.
WACCM periods are given below 147 yr from a model run of 150 yr. ECHAM6 periods are from a 400 yr run.
Short periods (below 20 yr) are not shown for WACCM, ECHAM6,  and GLOTI as they are not used in the
present paper. Hohenpeißenberg and GLOTI data after 1980 are not included in the analyses because of their
steep increase in later years.
Periods given in bold type are significant at the $1 - 2\,\sigma$  level or better, or are confirmed in the literature.

| No | HAMMONIA (119 layers) (years) | | WACCM (years) | | ECHAM6 (47 layers) (years) | | Hohenpeißenberg 1783 – 1980 (years) | | GLOTI 1880 - 1980 (years) | |
|---|---|---|---|---|---|---|---|---|---|---|
| 1 | **5.34** ± | 0.1 | | | | | 5.48 | ±0.21 | | |
| 2 | **6.56** | 0.24 | | | | | 6.16 | 0.20 | | |
| 3 | **7.76** | 0.29 | | | | | **7.83** | 0.26 | | |
| 4 | 9.21 | 0.53 | | | | | 9.50 | 0.65 | | |
| 5 | 10.8 | 0.34 | | | | | 10.85 | 0.38 | | |
| 6 | **13.4** | 0.68 | | | | | **13.6** | 0.80 | | |
| 7 | **17.3** | 1.05 | | | | | 18.02 | 1.08 | | |
| 8 | -- | -- | | | 20.0 | ± 0.35 | 19.9 | ± 1 | 20.2 | ± 1.36 |
| 9 | -- | -- | | | 20.9 | 0.15 | -- | -- | | |
| 10 | 22.8 | 1.27 | 21.7 | ± 1.02 | **22.1** | 0.23 | 21.9 | 0.94 | | |
| 11 | -- | -- | | | 23.8 | 0.42 | | | | |
| 12 | -- | -- | **25.82** | 0.86 | **25.3** | 0.46 | 25.1 | 0.62 | 25.5 | 2.0 |
| 13 | 28.5 | 1.63 | -- | -- | 27.3 | 0.41 | -- | -- | | |
| 14 | | | 31.56 | 1.42 | **30.2** | 0.49 | 29.8 | 0.66 | | |
| 15 | | | -- | -- | 33.3 | 0.84 | -- | -- | | |
| 16 | | | 38.1 | 0.82 | 36.9 | 1.17 | 36.01 | 1.28 | 35.4 | 2.42 |
| 17 | | | 41.89 | 0.95 | **41.4** | 0.97 | -- | -- | | |
| 18 | | | -- | -- | **48.4** | 1.73 | -- | -- | | |
| 19 | | | -- | -- | -- | -- | 52.06 | 1.61 | 53.4 | 11.4 |
| 20 | | | 57.64 | 1.69 | **58.3** | 1.77 | -- | -- | | |
| 21 | | | 66.95 | 7.31 | 64.9 | 2.98 | -- | -- | | |
| 22 | | | -- | -- | 77.5 | 3.94 | 81.6 | 4.18 | | |
| 23 | | | 97.27 | 5.06 | 95.5 | 5.86 | -- | -- | | |
| 24 | | | 147 | 14.9 | 129.4 | 14.5 | -- | -- | | |
| 25 | | | | | 206.7 | 16.3 | -- | -- | | |
| 26 | | | | | -- | -- | 238.2 | 11.8 | | |
| 27 | | | | | 341.2 | 37.2 | | | | |






Table 3
1329        Period comparison of two different HAMMONIA runs
Periods (in years) are given together with their standard deviations.
HAMMONIA run Hhi-max uses 119 altitude layers and covers 34 years; run Hlo-max uses 67 layers and covers
20 years.
No        Hhi-max        Hlo-max

| No | Hhi-max | | Hlo-max | |
|----|---------|------|---------|------|
| 1 | 2.06 ± 0.02 | | 2.07 ± 0.04 | |
| 2 | 2.16 | 0.02 | 2.15 | 0.02 |
| 3 | 2.33 | 0.04 | 2.36 | 0.03 |
| 4 | 2.51 | 0.04 | 2.43 | 0.02 |
| 5 | 2.79 | 0.08 | 2.78 | 0.07 |
| 6 | 3.11 | 0.08 | 3.2 | 0.09 |
| 7 | 3.52 | 0.12 | 3.44 | 0.15 |
| 8 | 3.96 | 0.08 | 3.9 | 0.12 |
| 9 | 4.48 | 0.21 | 4.27 | 0.21 |
| 10 | 5.34 | 0.1 | 5.48 | 0.29 |
| 11 | 6.56 | 0.24 | 6.57 | 0.29 |
| 12 | 7.76 | 0.29 | 8.02 | 0.12 |
| 13 | 9.21 | 0.53 | 9.16 | 0.33 |
| 14 | 10.8 | 0.34 | 11.05 | 0.46 |
| 15 | 13.4 | 0.68 | 13.02 | 0.83 |
| 16 | 17.3 | 1.05 | -- | -- |
| 17 | 22.8 | 1.27 | 22.68 | 1.11 |

1337  1
1338  2
1339  3
1340  4
1341  5
1342  6
1343  7
1344  8
1345  9
1346  10
1347  11
1348  12
1349  13
1350  14
1351  15
1352  16
1353  17





Table 4
Maxima / minima of accumulated amplitudes of temperature oscillations and
associated structures  (see Fig.11)
(stratosphere, mesosphere, lower thermosphere)

| altitude ( km ) | accumulated amplitudes | zonal wind | temperature gradient |
|---|---|---|---|
| 105 | max | westerly  (summer) | large  (positive) |
| 93 | min | westerly  (summer) | near zero |
| 84 | max | westerly  (summer) | large  (positive) |
| 78 | min | easterly  (except Sept) | medium  (negative) |
| 63 | max | westerly  (winter) | large  (negative) |
| 51 | min | westerly  (winter) | near zero |
| 42 | max | westerly  (winter) | large  (positive) |






Table 5
List of Acronyms

Acronym                    Definition

CCM                Chemistry Climate Model
CESM-WACCM         Community Earth System Model – Whole Atmosphere Community
1441                       Climate Model
ECHAM6             ECMWF/Hamburg
GLOTI              Global Land Ocean Temperature Index
HAMMONIA           HAMburg Model of the Neutral and Ionized Atmosphere
IPCC               Intergovernmental Panel on Climate Change
LOTI               Land Ocean Temperature Index