# Peer review of "DO 29.1.18 Vers3abcd Self-excited Oscillations in the Atmosphere (0 - 110 km)at Long Periods Dirk Offermann(1), Christoph Kalicinsky(1), Ralf Koppmann(1), and Johannes Wintel(1,2)Institut für Atmosphären - und Umweltforschung, Bergische Universität Wuppertal, Wuppertal, Germany Now at Elementar Anal"

_Atmospheric Chemistry and Physics, 2020_

## Author Comment (AC1) · 29 Jun 2020

Response to Anonymous Referee #1

Thank you for your review.

Your Comment: "Have you considered possible interpretation of your results in terms of stationary waves with vertical wavelength of 21 km (Fig.11)?" Reply: As we stated in Sections 4.1 and 5 of our paper we do not yet know the nature of the self-sustained oscillations. We avoided to interpret them in terms of "waves" for two reasons: 1) Typical atmospheric waves show an exponential amplitude increase with altitude, which is not the case for our oscillations (see Fig.1 and related text). 2) At the long periods discussed diabatic processes should be dominant. A possibility might be the modulation of atmospheric waves by the long period oscillations. This appears, however, to be a long shot. Furthermore, our oscillations appear to be influenced by changes in the (seasonal) wind direction. This would be difficult to understand for a stationary wave.

Your technical comment, Line 489: "intermittent" has been corrected. Thank you!

---

## Referee Comment (RC2) · Anonymous Referee #2 · 8 Jul 2020

**General comments:**

This manuscript deals with temporal oscillations in atmospheric temperature profiles obtained from simulations with three different global models (HAMMONIA, ECHAM6 and WACCM). The authors identify spectral signatures of "self-excited" oscillations in the model results and also a few other data sets. The paper is generally well written. I'm not really convinced all the findings and implications presented in this paper are robust or will turn out to be correct. For example, many of the spectral signatures listed in Table 2 are not significant, even at the 1-sigma level, if I interpret the table correctly. But several results are indeed interesting and somewhat stunning (e.g. the vertical structures in the correlations shown in Figure 3) and I think the paper should eventually be published to stimulate further research on this topic. In my opinion a

major revision is required before the manuscript should be accepted for publication in ACP.

Apart from the specific concerns listed below, I have the following general comments:

- Parts of the manuscript are not well organized. The Hohenpeißenberg and GLOTI data are not really introduced, but appear suddenly in the middle of the paper. I suggest adding sections with brief descriptions of these data sets to section 2. Also, section 2.1 (Self-sustained oscillations and their vertical structures) appears misplaced in section 2 (Model data and their analysis).

- The spectral analysis by subsequent fitting of different components is not a good technique, because it generally misses components with small amplitudes. This is a known problem with such an approach and I ask the authors to consider alternative methods not affected by this problem. This is probably also the reason, why signatures cannot be identified at certain altitudes in Fig. 1.

- If the observed effects are associated with vertical displacements (e.g. a shrinking and expanding atmosphere), then the results should differ depending on whether pressure or altitude is used as a vertical coordinate when analyzing the data. I suggest plotting the results as a function of both pressure and altitude to check for any systematic differences.

Specific comments:

Line 51: "Self-generated (self-sustained) oscillations"

Is it obvious that "self-generated" and "self-sustained" can be used as synonyms?

Line 139: "They are not linked to the ocean."

How can one know? If there is a reason, please state it.

Line 159: as mentioned above, section 2.1 does not really fit here.

Line 182: "At some altitudes the periods could not be determined."

Why not?

Figure 1 (related to the previous point): How can an amplitude and phase be provided, if the period is unknown? I don't understand how this is possible. Is there something wrong?

Line 222: "Here, the experiment with monthly varying constant"

Unclear, what "monthly varying constant" means. Please rephrase.

Line 226: "Solar cycle variability, however, was not kept constant"

I'm don't really understand what this means. The variability was not kept constant? Or do you simply mean that solar activity was not kept constant, i.e. varied?

Lines 290 – 294: Are the dominant spectral signatures the same for both models?

Line 311: "The HAMMONIA data used for Fig.4 were annual data that have been smoothed by a four point running mean"

What is the effect of the 4-point smoothing on the correlations? Some of the pronounced signatures have periods < 4 years.

Line 337 - 340: HAMMONIA and ECHAM are based on the same underlying model, right? How can the possibility be excluded, that the similarity in results is related to that? This aspect is discussed below and I suggest adding a brief reference to the following discussions.

Figure 6: I suggest adding an x-axis with periods in yr.

Line 418: "The Lomb-Scargle spectra (in their original form) do not reveal the phases of the oscillations."

Well, a Lomb-Scargle analysis will/can certainly also provide information on the phase.

Line 419 – 424: This doesn't seem like a good technique. You will potentially miss many features (see also general comment above).

Figure 7: I suggest adding a x-axis with periods and also using altitude rather than pressure as a vertical coordinate. If the main mechanism is related to vertical displacements (or shrinking & expansion of the atmosphere) it may make a difference, whether altitude or pressure is used as a vertical coordinate.

Line 446: "This procedure allows to obtain estimated amplitude and phase values for instance in the vicinity of the amplitude minima."

Well, one doesn't really know, how robust these estimates are, right? If I understand correctly, you assume a period, and then obtain an estimate of the amplitude and phase by fitting this period? Perhaps this can be mentioned more explicitly.

Line 458: "They suggest that the layer anti-correlation discussed above is at least in part due to the phase structure of the self-sustained oscillations in the atmosphere."

I don't think the logic behind this sentence is correct. The layer anti-correlation and the vertical phase structure are two manifestations of the same underlying phenomenon, right? I don't think it makes sense to state that one follows from the other.

Line 505: "Accumulated amplitudes have also been calculated for the ECHAM6 periods, and very similar results are obtained as for HAMMONIA."

I suggest also showing the results for ECHAM6.

Line 541/542: Here the Hohenpeißenberg and the GLOTI data appear suddenly and I suggest introducing them earlier, preferably in section 2.

Line 546: "the zero level data"

Please explain, what this means.

Line 569: "to 20 yr" -> "to 120 yr" ?

Line 608: "Two gradients are given for monthly mean temperature curves in addition:"

I don't quite understand this statement and why the two symbols appear at their specific altitudes. Please explain. Perhaps I'm missing a point here.

Legend Figure 13: perhaps the minus sign in "- delta T" can be omitted? It always the problem with the meteorological and the physics definition of the temperature gradient.

Lines 643 – 646: I suggest also showing the identified periods of the variations in CH4 in Table 2.

Line 651: "This means that the displacement mechanism is the same for all oscillations."

I'm not sure this conclusion is justified. Why should the displacement mechanism be the same, if the vertical displacement is essentially independent of the period? I don't necessarily see a direct connection between the two aspects.

Same sentence/paragraph: Please show a vertical profile of D (with scatter from the individual oscillations).

Also, if the mechanism is related to vertical displacements (shrinking and expansion of the atmosphere) then the results should look very different if analyzed on an altitude grid rather than a pressure grid, right? I suggest plotting the results both as a function of pressure and altitude and check, how they differ.

Line 785: Most of the signatures are not significant at the 1-sigma level, if I interpret the Table correctly. This questions your arguments a bit.

Figure 18: This is just a general comment, but I'm not asking for additional analysis: Wavelet transform would also be a good technique to investigate both time and spectral information. It should provide more robust results compared to the windowed FT performed here.

Line 815: "by means of harmonic analyses"

Please mention explicitly, what this means. FFT, Lomb-Scargle, wavelet transform etc. are all harmonic analysis methods in a general sense.

Line 850: "It needs to be emphasized that the oscillations discussed in the present paper are not influenced by the ocean as they occur even if the ocean boundaries are kept constant."

Well, they could still be influenced by the oceans to a certain extent, right? I would replace "influenced" by "caused".

Line 1287: "or are confirmed in the literature"

Please indicate, which ones are significant in your analysis and which ones appear in the literature (are they significant there?)

Table 2: If I interpret it correctly, then most spectral components are not significant at the 1 sigma level, right? This should be mentioned in the main text.

Also, please indicate – perhaps using color – which components are significant at the 1-sigma level, and which are significant at the 2-sigma level.

Typos etc.:

Line 16: Please add a space in "Atmosphären-und"

Line 73: "They are also seen in computer models (GCM) of the atmospheric."

Sentence is incomplete

Line 168: space missing in "et al., 2003)."

Line 264: "Figure 1" -> "Figure 1"

Line 268/269: Please delete space before comma.

Line 283: "Fig.3" -> "Fig. 3" (this occurs many times throughout the manuscript, please check the entire manuscript)

Line 290: "2.)" -> "2)" Line 355: "picrure" -> "picture"

Same line: "structur" -> "structure"

Line 489: "intermittend" -> "intermittent"

Line 540: "rows" -> "columns"

Line 705: "self- excitation" -> "self-excitation"

Line 821: "Table2" -> "Table 2"

Line 848: add space in "et al.(2010) with"

---

## Author Comment (AC2) · 27 Jul 2020

DO 8.7.20

Response to Anonymous Referee #2

We thank the Referee for his careful and detailed review. He states that the paper "...should eventually be published to stimulate further research...". This was exactly our intention, disregarding that the analysis was in part somewhat rough.

General Comments

1) "Parts of the manuscript are not well organized. The Hohenpeißenberg and GLOTI data are not really introduced, but appear suddenly in the middle of the paper. I suggest

adding sections with brief descriptions of these data sets to Section 2."

The Hohenpeißenberg and GLOTI data are now introduced early in the paper (Section 1, text in red).

" Also, Section 2.1 (Self-sustained oscillations and their vertical structures) appears misplaced in Section 2 (Model data and their analysis)."

This Section first sketches the results of earlier model analyses for shorter periods, in order to guide the reader to the present, longer period results. Secondly, it gives a basic picture of the self-excited oscillations (Fig.1) to allow the reader envisage the oscillations mentioned in the following text.

2) "The spectral analysis by subsequent fitting of different components is not a good technique, because it generally misses components with small amplitudes. This is a known problem with such an approach and I ask the authors to consider alternative methods not affected by this problem. This is probably also the reason, why signatures cannot be identified at certain altitudes in Fig.1.".

This problem was aware to us, as mentioned in Lines 449pp and 823. In Lines 423pp we stated that we show first approximations, only. To emphasize the point, we have now complemented the sentence in Line 424 as follows: "Furthermore, the 10% grid may be sometimes too coarse, and also small amplitude oscillations may be overlooked." In the present paper we want to give an overall picture of this type of oscillations, only, and therefore restrict ourselves to the larger and hence more important ones in Tab.2a.

3) "If the observed effects are associated with vertical displacements (e.g.a shrinking and expanding atmosphere), then the results should differ depending on whether pressure or altitude is used as a vertical coordinate when analyzing the data. I suggest plotting the results as a function of both pressure and altitude to check for any systematic differences."

The discussion of possible displacements has been much longer in the original version

of the manuscript, including several pictures. As this appeared to unduely increase the volume of the paper, we have limited this section to a short summay in Lines 642 – 656. We hope to give a more comprehensive presentation in a future paper.

Specific comments:

1) Line 51: "Is it obvious that "self-generated" and "self-sustained" can be used as synonyms?"

No!, you are right!, the two expressions are not synonymous. As our Section 3.6 ("Oscillation persistence") shows, the persistence of the oscillations may be limited. My dictionary tells me, however, that an "sustained oscillation" is one without damping. Thank you! We have now replaced "self-sustained" by "self-excited" in the text throughout.

2) Line 139 (Introduction): "They are not linked to the ocean." "How can you know? If there is a reason, please state it."

The self-excited oscillations are seen in the model results, even if the model boundary values concerning the ocean are kept constant. This is discussed in detail later in the paper. We have included a corresponding reference here: "They are not linked to the ocean (see below)."

3) Line 159: "as mentioned above, Section 2.1 does not really fit here"

See above, General comment #1.

4) Line 182: "At some altitudes the periods could not be determined. Why not?"

One reason may be too small an amplitude (see General comment 2 above). Other reasons are listed in Lines 548pp (for instance insufficient spectral resolution). We have added now a corresponding reference: ("... periods could not be termined (see Section 3.3).)

5) Figure 1 (related to the previous point): "How can an amplitude and phase be provided, if the period is unknown? I don't understand how this is possible.Is there somenone

thing wrong?"

The procedure is described in Section 2.1, Line 183pp: "In these cases the periods were prescribed by the mean...". The Levenberg-Marquardt algorithm allows this , as is described in Line 447pp. We have added a more detailed reference in Line 185: "Details are given in Section 3.2, Lines 447pp." In Line 449 we have replaced the "harmonic analysis algorithm" by "Levenberg-Marquardt algorithm".

6) Line 222: "Here, the experiment with monthly varying constant", "Unclear, what monthly varying constant means. Please rephrase."

It is meant that there is a seasonal variation, but it is the same in all years. A corresponding sentence has been added now.

7) Line 226: "Solar cycle variability, however was not kept constant". I don't really understand what this means. The variability was not kept constant? Or do you simply mean that the solar activity was not kept constant, i.e. varied?"

Yes, solar activity was varied, but in a special way: In the time interval 1955-2004 the measured data of Lean et al. were used. Thereafter the measured data from 1962-2004 were used as a proxy data block of 42 years length, to start at year 2005. At the end of this the block was repeated again and again until the total length of the time interval of 150 years had been reached. We have rephrased the paragraph accordingly.

8) Lines 290-294: "Are the dominant spectral signatures the same for both models?"

Yes! Please see Tab.2a!

9) Line 311: "The HAMMONIA data used for Fig.4 were annual data that have been smoothed by a four point running mean." "What is the effect of the 4-point smoothing on the correlations? Some of the pronounced signatures have periods smaller than 4 years."

The effect is unimportant. The altitude levels of maxima, minima, and zero crossings

are about unchanged. This is shown in Picture 1 attached to this "Response" for the HAMMONIA data. A corresponding sentence has been added to the text.

10) Line 337-340: "HAMMONIA and ECHAM are based on the same underlying model, right? How can the possibility be excluded, that the similarity in results is related to that? This aspect is discussed below and I suggest adding a brief reference to the following discussions."

a) Our analysis shows that the vertical correlation structure is due to the vertical phase structure of the self-excited oscillations (Section 3.2, Lines 458-459). These phase structures are similar for all oscillations. The periods are very similar for all models and even measured data (Tab.2a). It is therefore unlikely that the similarity of the HAMMONIA and ECHAM results is an artifact. b) Oscillation #10 in Tab.2a is the same in the HAMMONIA, WACCM, and ECHAM models. The WACCM model is quite different from the other two. c) Basically, we cannot answer the question of the referee from the data alone. We would have to go into the structure of the models, themselves, which is not possible to us. We, therefore, suggest to take the ECHAM data as an extension of HAMMONIA. We have added a corresponding remark in Section 2.4 (Lines 253pp).

11) Figure 6: "I suggest adding an x-axis with periods in years".

Was done as suggested.

12) Line 418: "The Lomb-Scargle spectra (in their original form) do not reveal the phases of the oscillations." "Well, a Lomb-Scargle analysis will/can certainly also provide informationon the phases",

The Lomb-Scargle algorithm we used ("original form") did not give phases.

13) Line 419-424: "This doesn't seem like a good technique. You will potentially miss many features (see also general comment above)."

See above reply to the General Comment #2.

14) Figure 7: "I suggest adding a x-axis with periods and also using altitude rather than pressure as a vertical coordinate. If the main mechanism is related to vertical displacements (or shrinking &expansion of the atmosphere) it may make a diffence, whether altitude or pressure is used as a vertical coordinate."

X-axis was complemented as suggested. Y-axis was left as it was given to us by the model provider because the vertical displacement was only sketched here (see above reply to General Comment #3).

15) Line 446: "This procedure allows to obtain estimated amplitude and phase values for instance in the vicinity of the amplitude minima." "Well, one doesn't really know , how robust these estimates are, right? If I understand correctly, you assume a period, and then obtain an estimate of the amplitude and phase by fitting this period? Perhaps this can be mentioned more explicitly."

The Levenberg-Marquard algorithm works as follows: An initial period is specified. The algorithm searches in the vicinity of it for a major period. It determines this period , its amplitude, and phase, including error bars. We have now added two corresponding sentences in Line 423pp: "The algorithm starts from a given initial period and looks for a major oscillation in its vicinity. For this it determines period, amplitude, and phase, including error bars. If in this paper the term "harmonic analysis " is used, this algorithm is always meant.

16) Line 458: "They suggest that the layer anti-correlation discussed above is at least in part due to the phase structure of the self-sustained oscillations in the atmosphere." " I don't think the logic behind this sentence is correct. The layer anti-correlation and the vertical phase structure are two manifestations of the same underlying phenomenon, right? I don't think it makes sense to state that one follows from the other."

This is a misunderstanding. I completely agree with you, and have therefore rephrased the sentence: ". . .the layer anti-correlation. . .corresponds . . .to the phase structure. . .".

17) Line 505: "Accumulated amplitudes have also been calculated for the ECHAM6 periods, and very similar results are obtained as for HAMMONIA." "I suggest also showing the results for ECHAM6."

Results for ECHAM6 are now shown in Fig.11b. The text has been modified accordingly.

18) Line 541/542: "Here the Hohenpeißenberg and the GLOTI data appear suddenly and I suggest introducing them earlier , preferably in Section 2."

See above reply to General Comment #1.

19) Line 546: "...the zero level data..." "please explain what this means."

We have now omitted these words as they are unnecessary.

20) Line 569: "to 20 yr...to 120 yr"

Has been corrected, thank you!

21) Line 608: "Two gradients are given for monthly mean temperature curves in addition" "I don't quite understand this statement and why the two symbols appear at their specific altitudes. Please explain. Perhaps I'm missing a point here."

The explanation was given in Lines 696pp (last two sentences of Section 3.5 (instead of the Figure legend of Fig.13).

22) Legend Figure 13: "Perhaps the minus sign in "-delta T" can be omitted? It is always the problem with the meteorological and physics definition of the temperature gradient."

The insert in Fig.13 has now been omitted as it contains redundant information, only.

23) Lines 643-646: "I suggest also showing the identified periods of the variations in CH4 in Table 2."

CH4 periods have now been added to Table 3 as suggested. The text has been complemented accordingly in Section 3.4.

24) Line 651: "This means that the displacement mechanism is the same for all oscillations." a) "I am not sure this conclusion is justified. Why should the displacement mechanism be the same, if the vertical displacement is essentially independent of the period? I don't necessarily see a direct connection between the two aspects."

This was a presumption, only. As mentioned, a detailed discussion is beyond the scope of this paper. We have modified the sentence accordingly: "This makes us presume that the displacement mechanism may be the same for all oscillations."

b) "Same sentence/paragraph: Please show the vertical profile of D (with scatter from the individual oscillations). Also, if the mechanism is related to vertical displacements (shrinking or expansion of the atmosphere) then the results should look very different if analyzed on an altitude grid rather than a pressure grid, right? I suggest plotting the results both as a function of pressure and altitude and check, how they differ."

Again, these detailed analyses are planned for a future paper. They would unduely increase the volume of the present paper.

25) Line 785: "Most of the signatures are not significant at the 1-sigma level, if I interpret the Table correctly. This questions your arguments a bit."

The discussion in Section 4.3a (Tab.2a) uses the periods obtained from the Levenberg-Marquardt algorithm with the corresponding error bars. These are 1-sigma errors.

26) Figure 18: Thank you for the comment.

27) Line 815: "by means of harmonic analyses" "Please mention explicitly, what this means. FFT, Lomb-Scargle, wavelet transform etc. are all harmonic analysis methods in a general sense."

We have complemented Line 815 accordingly: "by means of harmonic analyses (Levenberg-Marquardt algorithm)". We have also complemented the addendum to Line

446 (Comment #15) accordingly.

28) Line 850: "It needs to be emphasized that the oscillations discussed in the present paper are not influenced by the ocean as they occur even if the ocean boundaries are kept constant." "Well, they could still be influenced by the oceans to a certain extent, right? I would replace "influenced" by "caused".

Done as suggested!

29) Line 1287: "or are confirmed in the literature" "Please indicate, which ones are significant in your analysis and which ones appear in the literature (are they significant there?)"

A list of periods and their accuracies/significances has now been added as Table 2b. The text has been complemented accordingly. Some of the analysis methods and their accuracies are too complicated to discuss in this paper. The reader is referred to the original publication in these cases.

30) Table 2: "If I interpret it correctly, then most spectral components are not significant at the 1 sigma level, right? This should be mentioned in the main text. Also, please indicate – perhaps using color – which components are significacant at the 1 – sigma level, and which are significant at the 2-sigma level."

All periods derived from the Levenberg-Marquardt algorithm are significant at the 1-sigma level. This is now mentioned in the text (Section 3.3, 3rd paragraph). Significance levels are now explicitly shown in Tab. 2b.

Typos etc.:

Line 16: Please add a space in "Atmospären-und"

was added

Line 73: corrected to "They are also seen in computer models (GC) of the atmospheric." "Sentence is incomplete."

was corrected

Line 168: corrected

Line 264: corrected

Line 268/269: corrected

Line 283: "Fig. 3" "This occurs many times throughout the manuscript, please check the entire manuscript."

Manuscript was check as required.

Lines 290, 355, 489, 540, 705, 821, 848: corrected

Thank you for your efforts!

Picture 1 for #9
* * *
[Figure]

[Figure]

**Fig. 1.** picture 1for #9

---

## Referee Comment (RC3) · Anonymous Referee #3 · 30 Jul 2020

**General comments**

The authors present a statistical analysis of time series of annual temperature at a single place named "Central Europe" in three simulations. They present a number of periods, up to essentially the length of the data series, and claim that these are self-sustained oscillations of the atmosphere.

There are many points to be criticized in this study. First of all the claim of self-sustained atmospheric oscillations cannot be made, as the investigated simulations include also a land component that is coupled to the atmosphere and can provide memory for long time scales (cf. Hasselmann's Stochastic climate models, 1976). In many places in the text it would be best to remove the term self-sustained.

Further, oscillations with periods essentially the same as the length of the time series can certainly be identified in Fourier transforms, but the uncertainty in the spectral estimate for such periods is so large, that no claim on the existence of this period can be made.

A further weakness is the lack of discussion of the presented oscillations with respect to atmospheric dynamics. The vertical coupling of the stratosphere and troposphere or stratosphere and mesosphere in high and middle latitudes, in the winter/spring period, is an active field of research, but non of this is mentioned. Using this knowledge the study could have been focused on seasons from the beginning.

In the end this study is mainly a statistical time series analysis that identifies some periods in simulations made for "fixed climate conditions", including the atmosphere and land components. Some of the shorter periods have been reported by others for observational data.

Overall a major revision would be necessary to bring this manuscript in a publishable form.

**Specific comments**

L136: "... It is emphasized that on the contrary the self-excited multi-annual oscillations described by Offermann et al. (2015) and those discussed in the present paper are properties of the atmosphere, and exist in a large altitude regime between the ground and 110 km altitude. They are not linked to the ocean. ..."

It is a logical error to conclude from the occurrence of harmonic signals in atmospheric fields that these oscillation are of atmospheric origin. The reason is that models like HAMMONIA or WACCM couple the atmosphere to the land, through the hydrological cycle as well as the heat exchange. Thus more steps are needed to attribute the oscillation to the atmosphere alone. In the case of the QBO this has been clarified by the theory and a range of models from minimal 1-dimensional models to Earth system

**ACPD**
models.

L142: "... (Central Europe) ... "Please specify the place. Is it  $(45^{\circ}N-55^{\circ}N; 4^{\circ}W-16^{\circ}E)$  as in Offermann et al. (2015)? Please also explain why this place is chosen, and why only a single place is used. The model data are not limited to the single place. Would you expect the same results for a polar or equatorial place?

L143: "... The model boundary conditions (sun, ocean, trace gases) are kept constant. ... " Such model require more external data than sun (= spectral solar irradiance), ocean (surface temperature and sea ice concentration, at least), trace gases (CO2, CH4, N2O, CFCs): - Earth orbit parameters - natural and anthropogenic aerosol distributions in troposphere and stratosphere - land cover description and land use changes All of these contribute to low frequency variability. What was done with these sources of variability?

L162-165: "... These were found in temperature data of HAMMONIA model runs (see below). They were present in the model even if the model boundary conditions (solar irradiance, sea-surface temperatures and sea ice, boundary values of green-house gases) were kept constant. Therefore they were interpreted as self-sustained (self-excited) oscillations. ... "

It needs to be clarified if all external data were made constant-in-time. Further the text suggests that the presented oscillations are self-sustained (self-excited) oscillations of the atmosphere. But also this cannot be claimed as commented above. The text needs to be changed.

L167: "... Robust periods are typical of self-excited oscillations (Pikovsky et al.,2003) ... " But also for externally forced oscillations may be robust, e.g. diurnal and annual periods as a result of Earth rotation and orbit modulating the locally incoming SW flux at the top of the atmosphere. "Robustness" is also a relative term. The variance of ENSO indices has a spectrum of a certain width, such that no single robust period exists, but instead a the robustness of the spectrum could be discussed. Thus this
statement and citation, is problematic, as it is unclear whether this statement is correct for the types of oscillations described in this paper.

L175: "... whether such longer periods could also be self-excited in the models. ... " This should be broken up in two questions: (1) whether long periods can be identified in the simulations used here, and (2) which is the origin of any such oscillation.

L302-303: "Fig.2 HAMMONIA temperature residues at 0 km and 3 km altitude with fixed boundary conditions (see text)." Needs clarification. Is this at the "Central Europe" position?

L183-15: "... In these cases the periods were prescribed by the mean of the derived periods (dash-dotted red vertical line, 17.3 yr) to obtain approximate amplitudes and phases at these altitudes (see Offermann et al., 2015). ... " If no period can be determined, there is no value in determining phase and amplitude for a prescribed period. It is strongly recommended to remove these suggestive phase and amplitude data from this figure. The fact that this practice was used in the Offermann et al. (2015) article, does not justify to repeat this practice here.

L221-222: "... This 150 year run was analyzed from the ground up to 108 km. The model experiments are described in Hansen et al. (2014) ... " It is not clear which experiment is meant. The experiments listed in Hansen et al. (2014) are shorter than 150 years. Do you mean the experiment named "Fixed SSTs" listed there with a length of 56 years?

L254: "... A summary of the model properties is given in Table 1. ... " For boundary conditions, please add in Table 1 information for the other boundary conditions: orbit parameters, aerosol distributions, where prescribed: ozone, and land properties. For "ocean" boundary conditions, three different ways are used to express that ocean surface conditions are "climatological": - HAMMONIA: "SST fixed", - WACCM4: "climatological SST and sea ice", - ECHAM6: "fixed" Does HAMMONIA fix only SST, but not sea ice? Why is the ocean boundary condition for WACCM4 named "climatological", **ACPD**
while those for HAMMONIA and ECHAM6 are simply "fixed"? Is there a difference, or just different wording for the same?

L264-265: "... Figure1 indicates that there are some vertical correlation structures in the atmospheric temperatures. ... " This impression is evoked mainly by the phase profile, which includes many filled-in points at levels where not oscillation period could be determined, as shown by the gaps in the profile for the period.

L266: "Ground temperature" Does this mean surface temperature? Figure 2 shows in the legend for the black and red lines: 0 km, and 3 km, respectively. Are these heights above surface? (A land point would normally be higher than 0 km above sea level). Or are these heights above sea level? If so, how was the surface temperature extrapolated to sea level?

L269-270: "... The temperature fluctuations thus show the internal atmospheric variability ... " The soil model has its own prognostic temperature variable in a number of layers, so that the surface temperature in the end is determined not only by the atmospheric temperature, but also by the soil temperature. Therefore the variability in the surface temperature is not internal atmospheric variability, but rather internal variability of the coupled atmosphere – land system.

L286-27: "... with two maxima in the upper stratosphere ... " The maximum at 42 km height, where r=1, occurs by construction.

L311-313: "... The HAMMONIA data used for Fig.4 were annual data that have been smoothed by a four point running mean. This was done to reduce the influence of high frequency "noise" mentioned above, which is substantial (a factor of 2). ... " There are two problems with this figure. First of all, the similarity in heights of correlation and standard deviation peaks depends on the selected reference height for the correlation profile. If this was 30 km or 50 km instead of 42 km, the correlation profile would be different, and also their similarity to the standard deviation profile. Secondly, from the two profiles shown in Figure 4 one is based on annual data and the other on the
smoothed time series. This raises the question if the similarity pointed out is valid only for the smoothed standard deviation profile, but not for the suppressed time scale. To avoid this question, it would be necessary to show the standard deviation profile for annual data, even if more "noise" appears.

L332: "... The results are very similar to those of HAMMONIA ... " Here the same problems need to be addressed as for Figure 4.

L355: picrure  $\rightarrow$  picture

L360-361: "... Obviously, the computer simulations contain periodic temperature oscillations, ... " What seems obvious is potentially misleading, where periods are nearly as large as the time series. FFTs of the full time series alone do not give any information on the uncertainty. This can be provided by spectral estimators, which make use of (possibly partially overlapping) time windows. The use of time windows of course reduces the maximum period.

L362-364: "... Because the boundary conditions of the computer runs were kept constant, these oscillations cannot be excited from the outside. They are therefore interpreted as self-excited (self-sustained) oscillations, and thus as intrinsic properties of the atmosphere ... " The attribution of the variability to the atmosphere is wrong. It can be attributed to the atmosphere – land system, provided the boundary conditions are "fixed".

L378: "... The mean spectrum of all altitudes was determined ... " Is this simply the arithmetic average over all levels?

L386-387: "... For each representation we took noise from a Gaussian distribution ... " If the idea here is that "white noise" can be used as default, the question arises if the internal variability rather resembles white noise or red noise. If any redness is allowed for, the "2-sigma" line would would no longer be a horizontal line I Figure 8, but rather increase towards larger periods (smaller frequencies). Thus the main question

**ACPD**
here is why white noise is a good choice for estimating the background noise, against which oscillations should be identified. White noise has been assumed for instance by Hasselmann (1976) to represent forcing by weather, and to show that this would explain a red spectrum in the longer climate time scales. In the work presented here, the time scales are rather in the climate than in the weather time scale range, from which one could motivate the usage of red noise.

L397–398: "... A coupling mechanism between the layers has to be present to explain the observed mean Lomb-Scargle Periodogram for the ECHAM6 data. ... " Figure 8 collects by construction spectra from different levels into one graph, whether or not they are vertically coupled in the model. This cannot be derived from this graph alone, even if the coupling exists. A cross-spectral analysis between the levels of interest would be more suitable.

L438-450: "This analysis was performed for all altitude levels available. ... The harmonic analysis algorithm calculates an amplitude and phase if a prescribed (estimated) period is provided." I find it quite unfortunate that the authors use such techniques to derive amplitude and phase information for periods which cannot be determined in the preceding time series analysis. This generates a number of results which are highly questionable. Scientifically it would be much more rewarding to learn about the nature of the few frequencies which are strong, which can be identified in model simulations used here and in observational data.

L508-509: "... This is remarkable because many more oscillations are contained in the ECHAM6 data set than in HAMMONIA ... " This is to be expected because the underlying processes, which drive the dynamics of the models, are of the same nature. Then the shorter time scales existing in the shorter HAMMONIA simulation have to be expected also in the longer ECHAM simulation.

L535-536: "... The maximum period cannot be longer than the length of the computer run. ... " This holds for Fourier transforms. But spectral estimates must be limited to
considerably shorter periods, maybe half of the time series length or less, depending on the uncertainty one wants to allow for. Thus also the upper limit for periods (lower limit for frequencies) needs to be chosen smaller than the time series length.

L551-552: "... For the measured data in Table 2 it needs to be kept in mind that they were under the influence of varying boundary conditions. ... " Should the varying boundary condition - with respect to the atmosphere-land system of the model simulations – eliminate oscillations if these are self-sustained? I find this difficult to accept. Can you describe how this should work? But I would agree that the richer forcing of the real world, compared to the "fixed climate" model simulations, adds variance to the spectrum.

L627-629: "... Also shown in Fig.13 is the correlation profile of HAMMONIA from Fig.3 (black squares here). The two curves are surprisingly similar (correlation coefficient is 0.80. Outside the range shown the correspondence is lost.). ... " Here the correlation profile depends on the choice of the reference levels, and thus the reported similarity to the vertical temperature gradient profile also depends on the choice of the reference level. A change of the reference level from 42 km to 30 or 50 km would strongly modify the similarity and the conclusions drawn from this. Still, one can expect that the vertical expansion of certain oscillator modes will be confined by the general stratification of the atmosphere. But this needs to be shown differently.

L636-638: "... If an air parcel is displaced vertically by some distance D ("displacement height") a relative change in mixing ratio is observed ... " A vertical displacement alone will never change the composition of air. This happens only in the presence of chemistry or photo-chemistry, which is sensitive to for example temperature or radiative fluxes.

L728-730: "... An FFT analysis was performed in 12 equal time intervals (blocks of 32 yr length) in the altitude regime 0.01 - 1000 hPa and the period regime 2 - 40 yr. ... "Blocks of 32 years do not allow to compute amplitudes/phases for 40 year periods.
L878-879: "... and even if the boundary conditions of sun, ocean, and greenhouse gases are kept constant. ... " It is shown here only for "fixed" boundary conditions. Therefore it is better to write: "... where the boundary conditions ... "

---

## Referee Report (RR1)

Thank you for your reply, and for considering my review. I think that your article has results which are worth to be communicated. But I also feel that more modifications are still needed to avoid unnecessary irritations or misunderstandings. In the following I have summarized my main concerns, related to your responses.

Responses to general comments

*"The land component of the models has been kept constant, too. A corresponding remark was added in Section 2.2."*

I am sorry, but I could not find any sufficient explanation on the land component in section 2.2. Your remark that land parameters are fixed. But Schmidt et al. (2010), who describe the Hhi-max simulation, do not mention that the land component has been kept constant in this experiment. As they do not give any specific details on the land model configuration, it must be assumed that the land model is normally included and thus interacts with the atmosphere as in the underlying base model ECHAM5, for which Roeckner et al. (2003, 2006) are cited.
I think this is an important detail for the interpretation of the results and the conclusions which can be drawn. As long as this is not sorted out, you cannot claim that the oscillations observed in the atmosphere are self-sustained by the atmosphere basic dynamics. Still it is interesting to find such oscillation in simulations where no interactive ocean model is included.

*"Agreed! The longest periods in Tab.2a are shown just for completeness. They are not really used in the paper. The corresponding error bars in Tab.2a are large and thus are a warning. Nevertheless it is interesting to see that the longest periods of HAMMONIA and of WACCM find approximate counterparts within combined errors in ECHAM6."*

The explicit mentioning of the 341 year period in the key-points and abstract gives the message that you consider them as important enough to be highlighted. If you avoid this, the reader would not become disappointed when understanding later that the error bars are so large. It is certainly interesting enough to point out the multi-decadal time scales, which you diagnose in a system without an ocean component.

Responses to specific comments

*"This is a misunderstanding: We did not claim an atmospheric origin of the oscillations, but we said that the oscillations are atmospheric properties. We do not know yet the origin of the oscillations, as was stated several times in the paper. We certainly agree that clarification will presumably need a number of steps."*

If no claim is intended in an atmospheric origin of the oscillations, the wording needs to be adjusted in several places across the whole manuscript. I find it very irritating to read for instance in the Key Points: "self-sustained oscillations linked to the atmosphere basic dynamics" although you respond that you do not claim an atmospheric origin of the oscillations. "linked to atmosphere basic dynamics" in my understanding implies that the atmosphere is the cause.

---

## Editor Decision (ED1)

Editorial comment for paper

acp-2020-89

**Self-excited Oscillations in the Atmosphere (0 – 110 km) at Long Periods**

by D. Offermann et al.

I thank the authors for their replies and (small) changes to the manuscript. I am sorry to say that I am still confused about two essential aspects of this study, which I mentioned in the last round of revisions and which are still not properly addressed. I urge the authors to spend more time to improve these issues. In its current version, I cannot accept this paper for publication.

Major comments:

1) **Origin of oscillations.** In your last reply document, you write "This is a misunderstanding: We did not claim an atmospheric origin of the oscillations, but we said that the oscillations are atmospheric properties. We do not know yet the origin of the oscillations, as was stated several times in the paper." To me this is completely inconsistent with the paper. The title of the paper is "Self-excited oscillations in the atmosphere", which I cannot read differently than these are oscillations that are generated / excited by atmospheric processes! So how can you now say that you don't claim an atmospheric origin, I am very confused. If your reply reflects what you would like the reader to get from your study, then you must change the title, the abstract and many parts of the paper. Please also note that the first sentence of your conclusions reads "The structures analyzed in this paper are believed to be oscillations that are self-generated in the atmosphere." This is radically different from your last reply. This essential aspect must be made consistent from the title to the last line of the paper.

2) **The potential role of the land surface model.** There is still not enough specific information in the paper (nor in the reply document) for the reader to understand the setup of the land surface model used in your simulations. This point is important because in case of an interactive coupling of the atmosphere with the land surface model, this might be a potential cause for the oscillations. You write that "The land component of the models has been kept constant" but this is not clear enough. Most likely your land model has a seasonal cycle of vegetation and of soil moisture? Soil moisture is an important parameter here; is this variable interactively coupled to the atmosphere (as in normal GCMs) and therefore can there be interannual variations in soil moisture and land-atmosphere interactions? Did you maybe even use a dynamic vegetation model? These aspects are very important to clarify. I fully acknowledge that your expertise is not with the land model, but for this particular study more detailed information about this aspect of the model setup is essential.

A minor remark: In the short summary you write "However, a GCM can be changed arbitrarily!" Hopefully not. I think I understand what you like to say, but "arbitrary changes" sounds like unphysical model modifications.

---

## Author Response (AR2)

We thank again the Referee for his detailed comments.
Again, changes in our manuscript are marked in red, and replies to the Referee's
comments are given below *in italics*.

General Comments:

1.

*"The land component of the models has been kept constant, too. A corresponding remark was added*
*in Section 2.2."*

I am sorry, but I could not find any sufficient explanation on the land component in section 2.2. Your remark that land parameters are fixed. But Schmidt et al. (2010), who describe the Hhi-max simulation, do not mention that the land component has been kept constant in this experiment. As they do not give any specific details on the land model configuration, it must be assumed that the land model is normally included and thus interacts with the atmosphere as in the underlying base model ECHAM5, for which Roeckner et al. (2003, 2006) are cited.
I think this is an important detail for the interpretation of the results and the conclusions which can be drawn. As long as this is not sorted out, you cannot claim that the oscillations observed in the atmosphere are self-sustained by the atmosphere basic dynamics. Still it is interesting to find such oscillation in simulations where no interactive ocean model is included.

*a) There is still a misunderstanding: I do not believe that the oscillations are* caused *by the atmospheric* dynamics *. We do not yet know the origin! The misunderstanding obviously stems from the word "link" which in my mind means "connection, relation". I learn, however, that you  understand it as "origin, cause". I asked the colleagues around me and found that the understanding is divided. To avoid confusion, I now replaced "linked" by "related"  throughout the paper.*

*b) As concerns the land component, my model informant tells me: vegetation parameters (as leaf area, wood coverage) and ground albedo are kept constant. Other parameters are not (e.g.snow or ice on lakes ). It is believed that their influence on the oscillations is small, but it cannot be excluded. We have therefore included a corresponding paragraph to Section2.2:*

> *As concerns* the land parameters part of them were also kept constant (vegetation parameters as leaf area, wood coverage) and ground albedo. Others were not (e.g.snow and ice on lakes). Hence, some corresponding small (?) influence on our oscillations cannot be excluded.

*The text has been searched throughout to improve corresponding formulations.*

2.

*"Agreed! The longest periods in Tab.2a are shown just for completeness. They are not really used in the paper. The corresponding error bars in Tab.2a are large and thus are a warning. Nevertheless it is interesting to see that the longest periods of HAMMONIA and of WACCM find approximate counterparts within combined errors in ECHAM6."*

The explicit mentioning of the 341 year period in the key-points and abstract gives the message that you consider them as important enough to be highlighted. If you avoid this, the reader would not become disappointed when understanding later that the error bars are so large. It is certainly interesting enough to point out the multi-decadal time scales, which you diagnose in a system without an ocean component.

*The period of 341 yr has been omitted now from the key words, abstract, Tab.2a and the text throughout.*

Specific Comments

1.

*"This is a misunderstanding: We did not claim an atmospheric origin of the oscillations, but we said that the oscillations are atmospheric properties. We do not know yet the origin of the oscillations, as was stated several times in the paper. We certainly agree that clarification will presumably need a number of steps."*

If no claim is intended in an atmospheric origin of the oscillations, the wording needs to be adjusted in several places across the whole manuscript. I find it very irritating to read for instance in the Key Points: "self-sustained oscillations linked to the atmosphere basic dynamics" although you respond that you do not claim an atmospheric origin of the oscillations. "linked to atmosphere basic dynamics" in my understanding implies that the atmosphere is the cause.

*Please see above General Comment #1a.*

Response to Referee #2, (von Savigny)                              DO 28.9.20

We thank again the Referee for his detailed and careful comments.
We follow the line numbering used by the Referee. Again, changes in our manuscript are marked in red, and replies to the Referee's comments are given below *in italics*.

General Comments

1    There is, however, one major point that questions part of the results in my opinion. The periods listed in Table 2a are based on the harmonic fit approach as described in the text. The authors write in line 477: „The clusters are separated by major gaps, as is indicated by vertical dashed lines (black)."
Looking at the right parts of Fig. 9 and also Fig. 10 it's not really obvious what qualifies as a gap and what doesn't. Based on Figs. 9 and 10, the choice of gaps appears quite arbitrary. In turn, the derived „mean periods" within these clusters are also arbitrary. I argue that some of the periods determined are not really robust or may not be robust, because they depend on the choice of cluster bounds, which was done by you and not based on an objective approach. This does not appear to affect all identified periods, but probably a substantial part of them.

*We have modified the wording of Paragraph Lines 474 pp and added the subsequent Paragraph*:

*In determining the mean oscillation periods we have avoided subjective influences as follows: Periods obtained at various altitudes were plotted versus altitude as shown in Fig. 1 (middle column, red).When covering the period range 5 to 30 years nine vertical columns appeared. The definition criterion of the columns was that there should not be any overlap between adjacent columns. It turned out that such an attribution was possible. To make this visible we have plotted the histograms in Fig. 9 and 10. The pictures show that the column values form the clusters mentioned which are separated by gaps. The gaps that are the largest ones in the neighbourhood of a peak are used as boundaries (except at 7.15 yr). It turns out that if an oscillation value near to a boundary is tentatively shifted from one cluster to the neighbouring one the mean cluster values experiences only minor changes. Figure 10 shows that our procedure comes to its limits, however, for periods longer than 20 years (for HAMMONIA). This is seen in Tab.2a from the large error bars. We still include these values for illustration and completeness.*
    *It is important to note that all HAMMONIA values in Tab.2a (except 28.5 yr) agree with the Hohenpeißenberg values within the combined error bars. The Hohenpeißenberg data are ground values and hence not subject to our clustering procedure. Furthermore also all other model periods in Tab.2a have been derived by the same cluster procedure. The close agreement discussed in the text suggests that this technique is reliable.*

Specific Comments

1   Lines 278 and 282: the standard deviations are given with +-. Standard deviations
    cannot, however, be negative by definition.

    *Agreed! Text was corrected.*

2   Line 305: ´the significance is much better for ECHAM6).´

    Please show the 95% lines for ECHAM as well.

    *Done as required. Text was modified accordingly.*

*3*  Line 351: ´multiplied by 2)´

    I suggest mentioning briefly, why the values were multiplied by 2 (to improve clarity)

    *Multiplication by 2 was done for easier comparison to other curves. Text was
    complemented accordingly.*

4   Line 390 – 394: Did you scale the Gaussian noise in any way, e.g. to match the
    standard deviation of the temperature data?

    *As each Lomb-Scargle Periodogram is normalized with the variance of the noise in
    the same way as for the data (see Lines 385-387), the noise needs not to be scaled
    before.  A scaling of the noise to match the standard deviation or variance will not
    change the normalized power.*

5   Fig. 6: Period labels at top ´34 YR´ and ´4 YR´ not well placed.

    *Period scales have been improved in Fig. 6 and 7.*

6   Line 425: ´This was done by stepping through the period domain in steps 10% apart.´

    I wonder, how this assumptions affects the identified periods. How would the periods
    look like, if you had used 15% or 20% steps. Would you identify different periods?

    *We would not identify different periods, if the steps were chosen larger. However, in
    such a case certain period might be overlooked. This is the reason why we chose steps
    this narrow.*

7   Fig. 7: Period labels at top ´400 YR´ and ´10 YR´ not well placed.

*See above #5.*

8   Fig. 8: Please mention briefly in Fig. caption, what the red line shows.

*The straight red line is too complicated to explain in the caption. Reference to the text is made, instead.*

9   Line 477: ´The clusters are separated by major gaps, as is indicated by vertical dashed lines (black).´

As mentioned above, this seems to be a very arbitrary approach and the resulting main or mean periods will directly depend on you subjective choice of gaps.

*See reply to General Comment #1 above.*

10   Fig. 12: Please explain, what ´W´ means.

*Figure 12 was taken from the textbook of Schönwiese (1992, their Fig.57). The meaning of "w" is not explicitely explained in that text. However, from the context I conclude that "white noise" is meant.*

11  Fig. 13: The occurrence of the lower maxima (near 40 km) in the correlations is certainly not surprising, because at 42 km the correlation is an autocorrelation and the coefficient is 1. This means that you can directly choose the altitude, where a value of 1 occurs, but adjusting the reference altitude. It would be interesting to see how the Fig. changes if a different height is used as reference. Perhaps the 2 curves agree much better, if z = 35 km is used as a reference? I suggest showing 2 or 3 curves with different reference altitudes.

*A similar question has been raised by Referee #3 during the previous round. The corresponding answer should apply here , as well (o.k.?):*

" "a)   reference height:   No quantitative conclusions are drawn from the correlation profile. Text has been rephrased (Line 317).
The layered structures in question are also seen if a different reference altitude is chosen. This is shown for the altitudes suggested (30 km, 51 km) in Picture A below.""

[Figure]

*Picture A   Vertical correlation with reference altitudes 30 km, 42 km, 51 km.*
*HAMMONIA 38123,annual data ( unsmoothed).*

12   Line 667: ´ This is a relative change, only.´

I don't really understand this statement. This is an absolute change in mixing ratio, not a relative change (in %), right?

*This statement is a reply to a question of  Referee #3 in the last round. He had wondered whether an absolute photochemical change was meant, which is not the case.To hopefully clarify the point we have rephrased pp 667as follows:*

*"If an air column is displaced vertically by some distance D ("displacement height") a seeming change in mixing ratio is observed at a given altitude. This is a "relative" change, only, not a photochemical one.  It can be estimated by the product {D times mixing ratio gradient}."*

13    ´ Line 788:  ´The periods are robust, i.e. they are found with similar values in different models.´

 As pointed out above, I don't think all the periods are robust.

*Text has been modified as follows:* *Many of the periods appear to be robust, i.e. they are found with similar values….*

14  Line 798: ´Maxima of oscillation amplitudes appear to be associated with westerly (eastward) winds together with large temperature gradients (positive or negative). Amplitude minima are associated with either easterly (westward) winds or with near zero temperature gradients.´

This could be directly related to the propagation of planetary waves in a westerly wind regime (Charney-Drazin-criterion). I suggest discussing this briefly. PW are also associated with vertical displacments, which could explain some of the observed effects.

*We hesitate to believe that our oscillations are some kind of waves . This is because waves propagating in the atmosphere should show an exponential amplitude increase in the vertical direction. This is not the case here, as is shown in Fig.1, and is similar for all other oscillations. We therefore avoided the word "wave" in this paper.*

15  Line 849: ´ (b) The periods given in Table 2a were all calculated by means of harmonic analyses´

There's a logical error here. The use of the LM-algorithm and the potential occurrence of a common-mode failure cannot be used as an argument for non-spurious results, right!

*There appears to be a misunderstanding here. The Referee appears to read this text in connection to the previous paragraph (Lines 844-847). This was not our intention, but Lines 849 pp are a new paragraph "b" that does not discuss non-spurious results but leads to supplementary analyses.*

16  Typos etc.:

*Have been corrected. Thank you for the list!*

---

## Author Response (AR3)

DO 4.11.20

Response to Referee #3

Editorial comment for paper acp-2020-89 as of  30 Oct 2020

We thank the referee for his additional and interesting comments!
Again, changes in our manuscript are marked in red, and replies to the Referee's comments
are given below *in italics*.

General Comments:

We respond to #2 first:

2) **The potential role of the land surface model.** There is still not enough specific information in
the paper (nor in the reply document) for the reader to understand the setup of the land surface
model used in your simulations. This point is important because in case of an interactive coupling
of the atmosphere with the land surface model, this might be a potential cause for the oscillations.
You write that "The land component of the models has been kept constant" but this is not clear
enough. Most likely your land model has a seasonal cycle of vegetation and of soil moisture? Soil
moisture is an important parameter here; is this variable interactively coupled to the atmosphere
(as in normal GCMs) and therefore can there be interannual variations in soil moisture and landatmosphere
interactions? Did you maybe even use a dynamic vegetation model? These aspects are
very important to clarify. I fully acknowledge that your expertise is not with the land model, but
for this particular study more detailed information about this aspect of the model setup is essential.

*Thank you for detailing on this point! You are right:  we cannot exclude land surface
influences on our oscillations ( as yet)! We have therefore changed Title, Abstract, and text
throughout, as you suggest in #1. I hope that you will find these modifications satisfactory.
They are marked in red, see especially Section 2.2, 2^{nd} Paragraph, and Section 4.1, 1^{st}
Paragraph.*
*Section 2.2, 2^{nd} paragraph*
*…As concerns the land parameters,  part of them were also kept constant (vegetation*

*parameters as leaf area, wood coverage) and ground albedo. Others were not (e.g. snow and ice on lakes). Hence, some  influence on our oscillations cannot be excluded. We, therefore, put the expression "self-excited" in quotation marks in this paper.*

*Section 4.1, 1$^{st}$ paragraph….. Therefore they are supposed to be self-generated oscillations. However, as said in Section 2.2, some influence of land surface perameters cannot be excluded. A corresponding analysis is beyond the scope of this paper, though. and is planned for the future. As a reservation, the expression "self-excited" is used with quotation  marks in this text….*

*The essential point of our paper are the long-period oscillations and their various properties. Self-excitation is only part of them, and it is difficult to prove, indeed. We always said that we "suspect, suppose , interpret" it. As a warning for the reader we put "self-excitation" in quotation marks now.*

*As concerns land surface influences we have started to work upon them (in the Southern hemisphere). This is a lot of work, and beyond the scope of the present paper. It should be discussed in a future paper, together with the other future analyses needed to determine the nature of these oscillations. We say this in the text, Section 4.1 (see above) and Section 5:*

*Section 5. 3$^{rd}$ paragraph…. Land surface influences in addition need to be studied in the future.*

*First preliminary results of this intended work do not show essential land surface contributions. It is too early, however, to discuss this in this paper.*

1) **Origin of oscillations.** In your last reply document, you write "This is a misunderstanding: We
did not claim an atmospheric origin of the oscillations, but we said that the oscillations are
atmospheric properties. We do not know yet the origin of the oscillations, as was stated several
times in the paper." To me this is completely inconsistent with the paper. The title of the paper is
"Self-excited oscillations in the atmosphere", which I cannot read differently than these are
oscillations that are generated / excited by atmospheric processes! So how can you now say that
you don't claim an atmospheric origin, I am very confused. If your reply reflects what you would
like the reader to get from your study, then you must change the title, the abstract and many parts
of the paper. Please also note that the first sentence of your conclusions reads "The structures
analyzed in this paper are believed to be oscillations that are self-generated in the atmosphere."
This is radically different from your last reply. This essential aspect must be made consistent from

the title to the last line of the paper.

*I am really sorry that "in the atmosphere" versus "by the atmosphere" poses such a problem. I thought that this is a semantic problem, only, but maybe I am wrong as I am not a native English speaker. By "in the atmosphere" I meant something like "tides in the atmosphere" or "waves in the ocean". Neither of these are excited by the atmosphere or by the ocean.*
*Or another example from the atmosphere/ocean system and its complicated feedback processes: Is the AMOC (Atlantic Meridional Overturning Circulation) excited "in" the atmosphere/ocean system or "by" the atmosphere/ocean system? I would rather say: "in".*
 *I believe the solution lies in the word "feedback". If an oscillation is seen in a system that is caused by some feedback process one would not say "The oscillation is excited by the system", but "by the process in the system". Do you agree?*
 *Indeed, I am afraid that our oscillations are due to some feedback mechanisms. It will be part of the future analyses mentioned to clarify this! Corresponding sentences have been added to Section 4.1, 2ⁿᵈ Paragraph, and Section 5, 1ˢᵗ sentence:*

This may indicate three-dimensional atmospheric oscillation modes excited by some feedback mechanisms.

The atmospheric structures analyzed in this paper are supposed to be oscillations that are self-generated by some feedback mechanisms.

3) A minor remark: In the short summary you write "However, a GCM can be changed arbitrarily!"
Hopefully not. I think I understand what you like to say, but "arbitrary changes" sounds like unphysical model modifications

*We have now written "selectively" instead of "arbitrarily". Does this sound better to you?*

---

## Author Response (AR4)

DO 15.12.20

Response to the Editor  comments of 14.12.20

We thank the Editor for his suggestions and have modified the manuscript accordingly. Again, changes of the manuscript are marked in red. Replies th the Editor's comments are given below in italics.

Editor Comment:

Many thanks for your further revisions. I think there is one important issue remaining, it is the use of the term "self-induced" or "self-excited". As we briefly exchanged in German via Email, my main problem here is that your interesting oscillations are either spontaneously self-excited in (or by) the atmosphere or they are the result of some interaction and/or feedback with external forcing (e.g., land-surface processes). Since you cannot make a clear conclusion about the origin - as you mention - both options are possible, I strongly suggest that you omit the question of the origin from most parts of the paper and therefore also the term "self-excited". This does not weaken your study (in fact, I think it will make it much clearer), if you limit most parts of your paper to the presentation of the oscillations and their properties, and then only in the final part briefly discuss the different options of where these oscillations may come from. This will lead to a clearer and more elegant paper.

Please take your time to properly polish the text of your paper to achieve maximum clarity about presenting the solid results about the properties of these oscillations, and in the end, briefly discuss the more hypothetical part about their potential origin.

*Thank you for the suggestions! We have now modified the text accordingly, i.e have omitted all speculations about "self-excitation". Instead, we have added a short paragraph in Section 4.1 as follows*:

4.1  The oscillations  exist in computer models even if the model boundaries for the influences of the sun,  the ocean, the green house gases are kept constant. Therefore one might suspect that they are self-generated. The oscillation periods are robust, which is typical of self-excited oscillations. However, external excitation by land surface processes is a possibility.

*I hope that you will find these changes satisfactory*.